# *Aedes aegypti* Argonaute 2 controls arbovirus infection and host mortality

**Shengzhang Dong** [1] **& George Dimopoulos** [1] ✉

*Ae. aegypti* mosquitoes transmit some of the most important human viral diseases that are responsible for a significant public health burden worldwide. The small interfering RNA (siRNA) pathway is considered the major antiviral defense system in insects. Here we show that siRNA pathway disruption by CRISPR/Cas9-based *Ago2* knockout impaired the mosquitoes' ability to degrade arbovirus RNA leading to hyper-infection accompanied by cell lysis and tissue damage. *Ago2* disruption impaired DNA repair mechanisms and the autophagy pathway by altering histone abundance. This compromised DNA repair and removal of damaged cellular organelles and dysfunctional aggregates promoted mosquito death. We also report that hyper-infection of *Ago2* knockout mosquitoes stimulated a broad-spectrum antiviral immunity, including apoptosis, which may counteract infection. Taken together, our studies reveal novel roles for Ago2 in protecting mosquitoes from arbovirus infection and associated death.

The yellow fever mosquito *Aedes aegypti* is the principal vector of numerous medically important human pathogens, including dengue virus (DENV), Zika virus (ZIKV), chikungunya virus (CHIKV) and Mayaro virus (MAYV). These pathogens are transmitted between humans and mosquitoes and represent a major public health causing socio-economical burden globally[1,2]. The populations at risk of mosquito-borne diseases have grown substantially in recent years and could potentially extend to low-risk countries in Asia, Europe and North America because of the spread of *Aedes* mosquito vectors as a result of climate change[3–5]. Current methods to control mosquito-borne diseases that have emerged, or re-emerged, in many geographical regions are proving insufficient because of insecticide resistance and a lack of effective vaccines. Vector control remains the principal method for controlling and limiting mosquito-borne diseases[6]. In recent years, with the development of genome editing and mosquito transgenesis, novel control strategies based on population suppression or population replacement have been proposed as alternatives for the control of mosquito-borne diseases[7].

To defend against infection with arboviruses, mosquitoes use their innate immune system, which includes the small interfering RNA (siRNA) pathway, Toll pathway, JAK/STAT pathway, and other defenses[8,9]. These innate immunity pathways control systemic infection in mosquitoes by maintaining arbovirus infection intensity at a level that does not cause pathogenesis in the mosquitoes but still enables the transmission of the pathogens[10]. This balance is crucial for vector competence for arboviruses. Otherwise, mosquitoes would suffer a high fitness cost because of virus infection and likely evolve total refractoriness to infection, which would not enable the viruses to be transmitted. The siRNA pathway is the major antiviral defense system against arbovirus infections in *Ae. aegypti* and plays an important role in maintaining the mosquito-arbovirus balance[11,12]. The core components of the siRNA pathway include the RNase III enzyme Dicer 2 (Dcr2), a dsRNA-binding protein (dsRBP) called R2D2, and the endoribonuclease Argonaute 2 (Ago2)[13–15]. Double-stranded viral RNA is first recognized and digested by Dcr2 into 21 nt-siRNAs. These viral siRNAs are then incorporated into the RNA-induced silencing complex (RISC) with the integration of R2D2, leading to the degradation of the viral RNAs by Ago2.

*Ago2* is a member of the evolutionarily conserved *Argonaute* family, whose members feature typical PIWI and PAZ domains. In addition to the essential role of Ago2 in regulating gene silencing in the cytoplasm, this enzyme has been shown to regulate gene expression in the nucleus[16,17] and to play a role in translation repression and heterochromatinization[18]. Ago2 interacts with LaminB, a nuclear

[1]W. Harry Feinstone Department of Molecular Microbiology and Immunology, Bloomberg School of Public Health, Johns Hopkins University, 615 N. Wolfe Street, Baltimore, MD 21205-2179, USA. ✉e-mail: gdimopo1@jhu.edu

scaffolding protein, to form a nuclear complex and modulate chromatin topology in *Drosophila melanogaster*[19]. To date, functions of Ago2 beyond its involvement in the siRNA pathway have not been explored in any mosquito species.

In *Drosophila*, a defective siRNA pathway impairs the flies' ability to defend against viral infection, resulting in high mortality[14,15]. However, most data derived from experiments using *Drosophila*-virus models are based on artificial infections in which the virus is injected into the insect[14,20]. This injection approach does not reflect the natural infection cycle of arboviruses that are initiated through virus acquisition through an infected blood meal, after which the viruses infect the midgut tissue before being disseminated to other tissues[21]. Furthermore, the viruses used for *Drosophila* infection studies are different from those causing mosquito-borne human diseases such as dengue, Zika, yellow fever, Mayaro, and chikungunya[22]. Therefore, to study the role of the siRNA pathway in controlling arbovirus infection in mosquitoes, we have now generated siRNA pathway-defective mosquitoes by knocking out *Ago2* in *Ae. aegypti*. Our study shows a fundamental role for Ago2 and the siRNA pathway in controlling arbovirus infection in mosquitoes, as well as in protecting the mosquitoes against mortality upon arboviral infection by modulating DNA repair, apoptosis, and autophagy. Our study provides insights into the molecular interactions between mosquitoes and arboviruses that may facilitate the development of novel disease control strategies.

## Results

### Generation and characterization of *Ago2* knockout mutants

To demonstrate the essential function of Ago2 in the *Ae. aegypti* siRNA pathway, we used CRISPR/Cas9 to knock out *Ago2* and then investigated its role in defending against arbovirus infection (Fig. 1a). *Ae. aegypti Ago2* (*AeAgo2*) has four exons and four predicated functional domains, and its PAZ and Piwi_Ago-like domains are considered essential for endonuclease activity (Fig. 1b and Supplementary Fig. 1a). Therefore, we designed guide RNAs (gRNAs) targeting the Piwi_Ago-like domain encoded by exon 3. Meanwhile, another two gRNAs targeting the ArgoN domain encoded by exon 1 were designed and used to induce frameshift mutations that resulted in *Ago2* disruption. Through embryo injection and leg PCR screening, we generated at least two independent *Ago2* knockout mutant lines: one line (*ArgoN*$^{-/-}$) with a mutation in exon 1 and another line (*Ago2*$^{-/-}$) with a mutation in exon 3 (Fig. 1c and Supplementary Fig. 1b). *ArgoN*$^{-/-}$ has a frameshift mutation (a 51-bp deletion and 14-bp insertion) in the first reading frame but uses the second reading frame to translate a truncated protein (Supplementary Fig. 1c). *Ago2*$^{-/-}$ line has an in-frame mutation with a 72-bp deletion in the Piwi_Ago-like domain.

To assess whether the siRNA pathway was impaired in *Ago2* knockout mutant lines, we performed three independent assays of RNA interference-based silencing different endogenous genes in WT and *Ago2* knockout mutants. (1) We attempted silencing of the *kynurenine hydroxylase* (*kh*) gene (AAEL008879) in embryos of WT and *Ago2*$^{-/-}$ and *AgoN*$^{-/-}$ mutants through microinjection of *kh* dsRNA. The *kh* gene is essential for the kynurenine 3-monoxygenase (KMO) function which is critical for eye pigmentation, and knockout of the *kh* gene leads to severe eye pigmentation defects[23]. As compared to the GFP dsRNA-injected control, a serious defect in eye pigmentation was observed in 3rd instar larvae hatched from *kh* dsRNA-injected WT eggs, however, we did not observe any defect in eye pigmentation in larvae hatched from *kh* dsRNA-injected *Ago2*$^{-/-}$ eggs (Supplementary Fig. 1d). We observed a mild defect in eye pigmentation in larvae hatched from *kh* dsRNA-injected *AgoN*$^{-/-}$ eggs. These results indicated that the *Ago2*$^{-/-}$ mutant is completely defective of the siRNA function, therefore a *Ago2* null mutant, while the *AgoN*$^{-/-}$ mutant still has some residual siRNA function, therefore a partially *Ago2* knockout mutant. (2) We attempted silencing of the *eggshell organizing factor 1* (*EOF1*) gene (AAEL012336) in WT and *Ago2*$^{-/-}$ females through injection of

*EOF1 dsRNA*. *EOF1* plays an essential role in the formation and melanization of the eggshell and silencing of *EOF1* leads to a defect in eggshell melanization and reduction in fecundity[24]. *EOF1* dsRNA-injected WT females produced significantly less eggs and eggs showed defects in eggshell melanization when compared to the GFP dsRNA-injected control, however, *EOF1* dsRNA-injected *Ago2*$^{-/-}$ females did not show any difference in fecundity and eggshell melanizaiton as compared to the GFP dsRNA-injected controls (Supplementary Fig. 1e). (3) We attempted silencing of three genes (*AeIMPDH*, *AeCRVP* and *AeTry196*) that had been reported to show significant silencing through dsRNA injection in our previous studies[25,26]. Compared to WT where gene silencing was significant, *Ago2*$^{-/-}$ mutants didn't show gene silencing (Supplementary Fig. 1f). Taken together, our data demonstrate that RNAi-mediated silencing is fully abolished in the *Ago2*$^{-/-}$ mutants.

Multiple fitness parameters were assayed in WT and *Ago2*$^{-/-}$ mutants. The results showed that there were no differences in whole-body weight (Supplementary Fig. 2a), amount of ingested blood (Supplementary Fig. 2b), percentage of blood-fed mosquitoes (Supplementary Fig. 2c) and fertility of eggs between the WT and *Ago2*$^{-/-}$ mutants (Supplementary Fig. 2f). However, the *Ago2*$^{-/-}$ mutants have a significantly delayed larva development (Supplementary Fig. 2d), a reduced fecundity (Supplementary Fig. 2e) and a shorter life span of females and males (Supplementary Fig. 2g).

### Ago2 is required for controlling arboviral infection in mosquitoes, and *Ago2* disruption reduces arbovirus transmission

We performed independent natural infections of various *Ago2* knockout mutant lines and the control (the parental *Cas9* line) mosquitoes with different arboviruses, including two flaviviruses (DENV2 and ZIKV) and one alphavirus (MAYV) through blood meal. As compared to the control *Cas9* mosquitoes, both *ArgoN*$^{-/-}$ and *Ago2*$^{-/-}$ mutants exhibited a significantly higher titer of each of the three tested viruses, when measured at 14 days post-infection (dpi) for DENV2 and ZIKV and at 7 dpi for MAYV (Fig. 1d). At earlier time points, the virus titer was not significantly different between *ArgoN*$^{-/-}$ mutants and the control mosquitoes: at 7 dpi in the case of DENV2 and 3 dpi in the case of MAYV, however, the virus titer in *Ago2*$^{-/-}$ mutants was significantly higher than that of the *Cas9* controls at both the early and late time points for all three viruses. Hence, a mutation in the Piwi_ago-like domain had the strongest impact on arbovirus infection and Ago2 endonuclease activity, which was also shown in dsRNA-mediated silencing of endogenous genes (Supplementary Fig. 1d), and the *Ago2*$^{-/-}$ line was therefore used as a *Ago2* mutant line for the subsequent experiments. Midguts of *Ago2*$^{-/-}$ females displayed a different infection pattern upon arbovirus infection: for MAYV at 4 dpi, the infection of the control midgut was restricted to specific areas, whereas half the *Ago2*$^{-/-}$ midgut was heavily infected (Fig. 1e). At 7 dpi, the whole midgut of *Ago2*$^{-/-}$ females was strongly infected by MAYV, and the midguts also assumed a rounder shape than those of the *Cas9* midguts. At 7 dpi and 14 dpi with DENV2, a more intense infection was observed in midguts of *Ago2*$^{-/-}$ females when compared to those of the *Cas9* midguts (Fig. 1f). The tracheas associated with midguts were heavily infected by MAYV or DENV2 in *Ago2*$^{-/-}$ mutants when compared to the control mosquitoes (Supplementary Fig. 3). These data demonstrate that Ago2 plays an essential role in controlling virus infection intensity in *Ae. aegypti* and mosquitoes become hypersensitive to arbovirus infection when *Ago2* is disrupted.

To determine the effect of *Ago2* disruption on the transmission potential of arboviruses, we examined the levels of virus particles released into the feeding solution from the proboscises of arboviruses-infected *Ago2*$^{-/-}$ and WT females through an artificial feeding system. As compared to the WT females, *Ago2*$^{-/-}$ females exhibited a significant reduction of virus particles released into the feeding solution at 14 dpi with DENV2 or 5 dpi with MAYV, and only 4 of 15 feeding solution

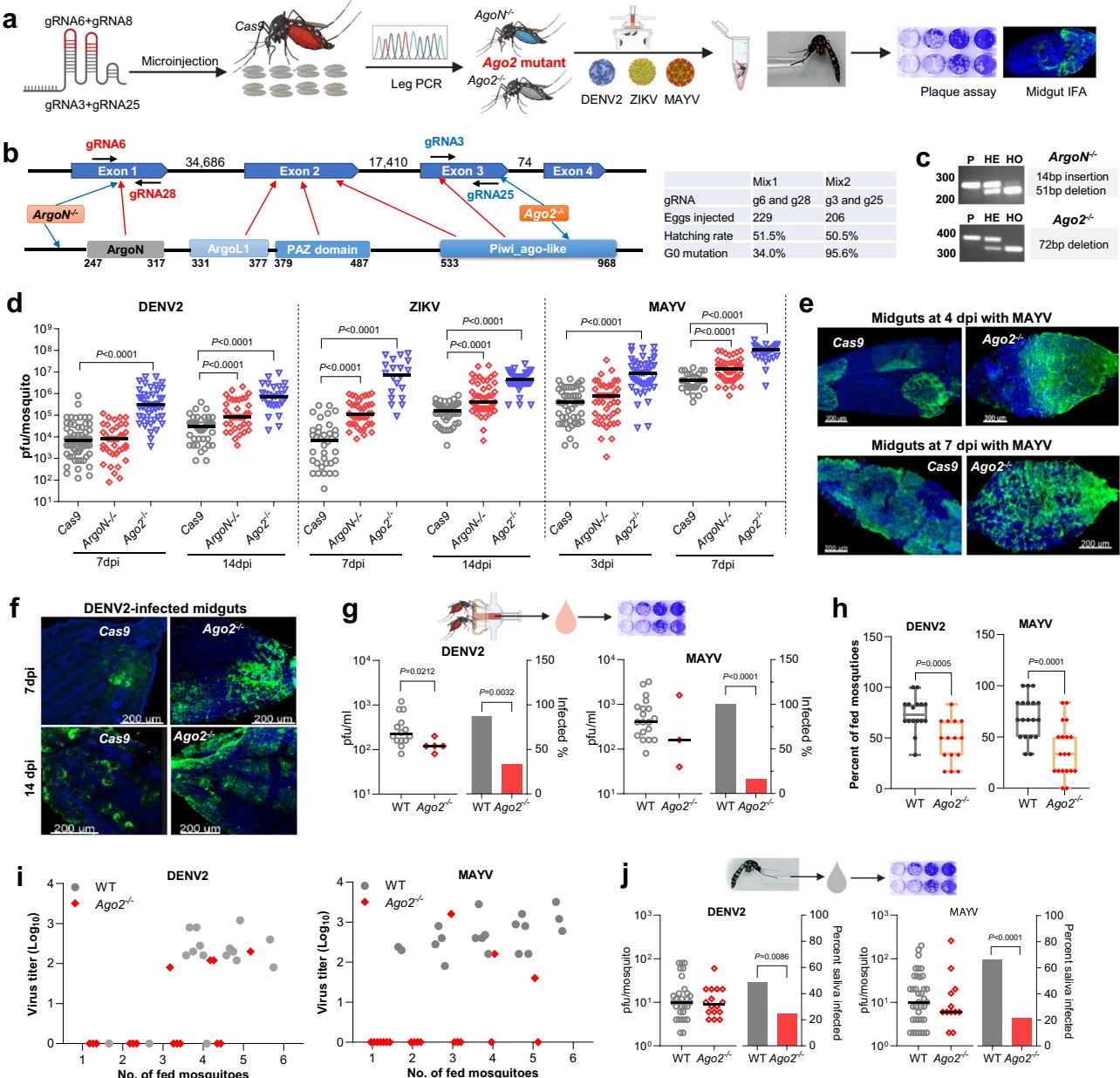

**Fig. 1 | Generation of *Ago2* knockout *Ae. aegypti* lines and the effect of *Ago2* disruption on arbovirus infection and transmission. a** flowchart of experiment design. **b** gene structure of *Ago2*, predicted functional domains, and guide RNA (gRNA) design for CRISPR/Cas9. The table shows data from generating *Ago2* knockout lines. **c** *Ago2* amplification in *Ago2* knockout lines. P, parental line *Cas9*; HE, heterozygous mutants; HO, homozygous mutants. **d** virus titer in *Ago2* knockout (*Ago2⁻/⁻* and *ArgoN⁻/⁻*) and *Cas9* mosquitoes on various days post-infection (dpi) with DENV2, ZIKV, or MAYV. IFA detection of MAYV (**e**) and DENV2 (**f**) antigen in midguts of *Cas9* and *Ago2⁻/⁻* mosquitoes at different days post virus infection. MAYV and DENV2 were detected with the corresponding monoclonal antibody (green). Nuclei were stained with DAPI (blue). **g** virus titer and infection prevalence in the feeding solution collected from a feeder exposed to a group of females at 14 dpi with DENV2 or 5 dpi with MAYV. The Liverpool strain was used as a control (WT). For DENV2 infection, *n* = 16 for WT and *n* = 15 for *Ago2⁻/⁻*; for MAYV

infection, *n* = 19 for both WT and *Ago2⁻/⁻*. **h** percentage of the fed mosquitoes in each group of females at 14 dpi with DENV2 or 5 dpi with MAYV. Data were presented as box and whiskers (Min to Max). **i** virus titer plotted against the number of the fed mosquitoes in each cup. **j** virus titer and infection prevalence of individual saliva samples collected from *Ago2⁻/⁻* and WT at 14 dpi with DENV2 or 5 dpi with MAYV. For DENV2 infection, *n* = 62 for WT and *n* = 61 for *Ago2⁻/⁻*; for MAYV infection, *n* = 56 for both WT and *Ago2⁻/⁻*. Each experiment comprised at least two biological replicates, and the data were pooled for generating the graphs. Virus titers were determined by plaque assay, and horizontal lines indicate the medians of the virus titer (**d**, **g** and **j**). *P* values were determined by an unpaired two-sided Mann-Whitney test for virus titer, an unpaired two-sided Fisher's exact test for infection prevalence, or an unpaired two-sided *t*-test for (**h**). Source data are provided as a Source Data file.

samples collected from the feeders were positive for DENV2 and 3 of 19 were positive for MAYV after exposure to *Ago2⁻/⁻* females, but 14 of 16 feeding solution samples were positive for DENV2 (Fig. 1g). In contrast, all feeding solution samples exposed to infected WT females were positive for MAYV (Fig. 1g). We also investigated whether arbovirus-infected *Ago2⁻/⁻* females displayed any altered feeding propensity (the

portion of the fed mosquitoes), and we found that the number of either DENV2- or MAYV-infected *Ago2⁻/⁻* females that had fed was significantly lower than that of the WT mosquitoes (Fig. 1h). We could not detect any DENV2 or MAYV in the feeding solution of some groups of virus-infected *Ago2⁻/⁻* mosquitoes that were fed (Fig. 1i), suggesting that some *Ago2⁻/⁻* mutants may fail to release virus into their saliva. To

further examine this, we used a forced-salivation method to compare the number of viral particles released in the saliva of *Ago2^-/-* and WT females at 14 dpi with DENV2 or 5 dpi with MAYV. We found no significant difference in the virus titer in the saliva samples from *Ago2^-/-* mutants and WT mosquitoes for both DENV2 and MAYV, but the *Ago2^-/-* mutants showed a significantly reduced virus prevalence (i.e., a reduced proportion of mosquitoes having virus particles in the saliva; Fig. 1j). Taken together, these data indicate that *Ago2* disruption significantly reduces the arbovirus transmission potential of mosquitoes by affecting the feeding propensity and the shedding of virus particles into the saliva from the salivary glands.

## Ago2 is not required for small RNA biogenesis in mosquitoes, but Ago2 disruption impairs viral RNA degradation

To determine whether the effect of *Ago2* disruption on arbovirus infection was the result of a defective siRNA pathway, we performed

small RNA (sRNA) sequencing of *Ago2^-/-* mutants and WT mosquitoes after MAYV infection. At 4 dpi with MAYV, a significantly higher number of MAYV sRNA was observed in *Ago2^-/-* mutants as compared to the WT mosquitoes (Fig. 2a). qPCR analyses showed that the MAYV titer in *Ago2^-/-* mutants was significantly higher than that in the WT mosquitoes (Supplementary Fig. 4a), and a principal component analysis (PCA) indicated that the high yield of vsRNAs could be explained by the high virus titer in *Ago2^-/-* mutants (Fig. 2b). The length of the viral sRNAs (vsRNAs) was distributed between 18 nt and 40 nt with a dominant peak at 21 nt at 4 dpi with MAYV in both *Ago2^-/-* mutants and WT mosquitoes (Fig. 2c), demonstrating that *Ago2^-/-* mutants still produced 21 nt-biased vsRNAs. *Ago2^-/-* mutants also produced a number of non 21-nt vsRNAs (Fig. 2c), and these vsRNAs had a prominent positive-strand bias (Supplementary Fig. 4b).

Since the biogenesis of vsRNA was not affected by *Ago2* disruption, we investigated whether the number of 21 nt viral small

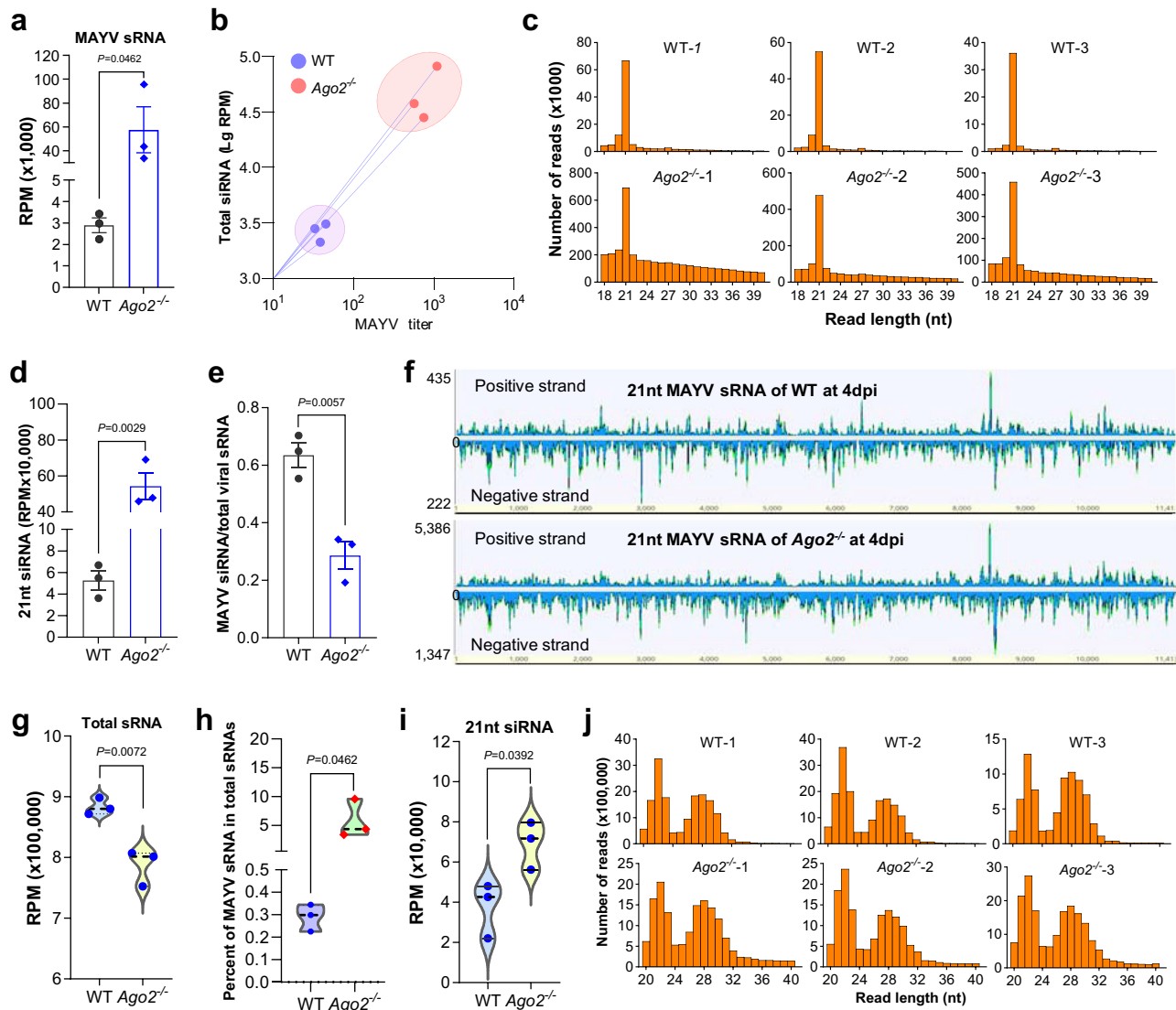

**Fig. 2 | Impact of *Ago2* disruption on production of MAYV- and mosquito-derived small RNAs. a** total number (reads per million mapped reads, RPM) of MAYV sRNAs in *Ago2^-/-* mutants and WT mosquitoes at 4 days post-infection (dpi) as determined by sRNA sequencing. Three replicates were performed for each sample. **b** correlation between the abundance of sRNAs and titer of MAYV in *Ago2^-/-* mutants and WT mosquitoes as determined by principal component analysis (PCA). **c** distribution by length of sRNAs mapping to the MAYV genome. **d** number of 21-nt siRNAs mapped to the MAYV genome. **e** the ratio of siRNA-sized (21 nt) MAYV reads

to the total MAYV sRNA (18nt to 40nt) reads. **f** distribution and the relative abundances of 21-nt MAYV siRNAs in the MAYV genome. **g** number of total sRNAs mapped to the *Ae. aegypti* genome. **h** percent of MAYV sRNAs in the total sRNAs in mosquitoes. **i** number of 21-nt siRNAs mapped to the *Ae. aegypti* genome. **j** length distribution of the mosquito sRNAs. *P* values were determined by using an unpaired two-sided *t*-test (*n* = 3) with GraphPad Prism. Data are presented as mean ± SEM (**a**, **d** and **e**). Source data are provided as a Source Data file.

interfering RNAs (vsiRNAs) that bound to siRNA pathway factors was altered in *Ago2*$^{-/-}$ mutants. A higher number of MAYV vsiRNA was observed in *Ago2*$^{-/-}$ mutants as compared to WT mosquitoes (Fig. 2d), but the ratio of vsiRNA reads to the total vsRNA reads was lower in *Ago2*$^{-/-}$ mutants (Fig. 2e) because they produced a greater number of non-21nt-vsRNA as compared to WT mosquitoes (Fig. 2b). There was a similar distribution of 21 nt vsiRNA in the positive or negative strand of the MAYV genome between *Ago2*$^{-/-}$ mutants and WT mosquitoes, but the relative abundances of the vsiRNAs were different because of the higher vsRNA reads in *Ago2*$^{-/-}$ mutants (Fig. 2f). Therefore, our results show that *Ago2* disruption does not affect the production of arboviral sRNAs but does impair the ability of the siRNA pathway to degrade viral RNAs, as reflected by a high virus titer and an impaired RNA silencing in *Ago2*$^{-/-}$ mutants.

The total sRNAs were also mapped to the *Ae. aegypti* genome. The results showed that *Ago2* disruption significantly reduced the abundance of the mosquito sRNAs at 4 dpi with MAYV (Fig. 2g), likely because of a higher fraction of MAYV sRNAs in the total sRNAs in *Ago2*$^{-/-}$ mutants (Fig. 2h). As compared to WT mosquitoes, *Ago2*$^{-/-}$ mutants produced a higher number of mosquito 21 nt siRNAs (Fig. 2i), suggesting that *Ago2* disruption may increase the abundance of the mosquito endogenous dsRNAs. The length distribution of the mosquito sRNAs showed no difference between *Ago2*$^{-/-}$ mutants and WT mosquitoes, and both showed two peaks at 21–23 and 26–30 nt, corresponding to microRNAs and piwiRNAs, respectively (Fig. 2j).

To further determine the influence of the *Ago2* disruption on the mosquito's ability to degrade arbovirus RNA, we performed independent sRNA sequencing on another *Ago2* mutant line (*ArgoN*$^{-/-}$) and its parental line (*Cas9*) at 2 dpi and 7 dpi with DENV2. The sRNA sequencing results of DENV2-infected *ArgoN*$^{-/-}$ mutants were very similar to what we observed in MAYV-infected *Ago2*$^{-/-}$ mutants, including a significantly higher number of total DENV2 sRNA (Supplementary Fig. 4c, d, e), 21nt DENV2 siRNA (Supplementary Fig. 4f, g), total mosquito sRNA (Supplementary Fig. 4h), and 21 nt mosquito siRNA reads (Supplementary Fig. 4i), and no difference in length distribution of the mosquito sRNAs (Supplementary Fig. 4j) in *ArgoN*$^{-/-}$ mutants as compared to the *Cas9* mosquitoes, and the high yield of DENV2 vsRNAs could be resulted from the high virus titer in *ArgoN*$^{-/-}$ mutants (Supplementary Fig. 4k).

Overall, our studies have shown that *Ago2* disruption does not affect the biogenesis of sRNAs in either arboviruses or mosquitoes and the increased abundance of viral sRNAs that we found is due to the high arbovirus titer in *Ago2*-deficient mosquitoes.

## Arbovirus infection results in high mortality of *Ago2*-deficient mosquitoes through cell lysis

To determine whether hyper-infection of arboviruses can lead to the death of *Ago2*$^{-/-}$ mutants, we performed a mortality assay involving DENV2, ZIKV, and MAYV infections transmitted through oral blood-feeding as well as through direct injection of viruses into the mosquito hemolymph. Our results showed that *Ago2*$^{-/-}$ mutants experienced a significantly higher mortality upon infection with the three different arboviruses than did WT mosquitoes infected through the two different infection routes, but direct injection of the viruses resulted in higher mortality than did natural infection through blood-feeding (Fig. 3a). Of all the arboviruses we tested, MAYV infection resulted in the highest mortality of *Ago2*$^{-/-}$ mutants (Supplementary Fig. 5a), and the median survival after MAYV infection was 6 days for the orally infected and 4 days for the virus-injected *Ago2*$^{-/-}$ mutants (Fig. 3a); therefore, *Ago2*$^{-/-}$ mutants and MAYV infection were used for the future experiments to explore the mechanism of arbovirus-induced mortality of *Ago2*$^{-/-}$ mutants. Arbovirus infection also caused the mortality of *ArgoN*$^{-/-}$ mutants, a mutation in the *ArgoN* domain of Ago2, but it is lower than that of *Ago2*$^{-/-}$ mutants (Supplementary Fig. 5b); these results were correlated with the virus titer in the various *Ago2*

knockout mutant lines (Fig. 1d), suggesting that the high mortality of the *Ago2*$^{-/-}$ mutants may be a result of their high infection intensity. The virus infection intensity-dependent mortality was then confirmed through injection of different titers of MAYV into *Ago2*$^{-/-}$ mutants where a higher titer of MAYV led to higher mortality of *Ago2*$^{-/-}$ mutants (Supplementary Fig. 5c).

We also used IFA to examine the infection of the midguts and salivary glands of *Ago2*$^{-/-}$ mutants and WT mosquitoes after MAYV and DENV2 infection. At 4 dpi with MAYV, the midguts of the *Ago2*$^{-/-}$ females showed a strong fluorescence with a "ball"-like shape that comprises hyper-infected midgut cells, indicating that the insects were hyper-infected by viruses (Fig. 3b), and the nuclei of a large proportion of the midgut cells were highly condensed and exhibited an abnormal shape when compared to those of the WT midguts (Fig. 3b); these findings suggested that the cells of *Ago2*$^{-/-}$ mutants were likely undergoing lysis as a consequence of virus hyper-infection. At 7 dpi with MAYV, the infection and lysis of midgut epithelial cells were even higher in *Ago2*$^{-/-}$ females, and most of the midgut cells were lysed (Fig. 3b). At 7 dpi and 14 dpi with DENV2, the higher infection and lysis of the midgut epithelial cells were also observed in *Ago2*$^{-/-}$ females when compared to the WT midguts (Supplementary Fig. 5d). A similar pattern of condensed nuclei and hyper-infection was observed in the salivary glands of *Ago2*$^{-/-}$ females upon MAYV infection (Fig. 3c), suggesting that cell lysis might occur in multiple tissue and cell types in *Ago2*$^{-/-}$ mutants in response to arbovirus infection. We did not observe lysis of midguts and salivary glands in uninfected *Ago2*$^{-/-}$ mutants as compared to the WT mosquitoes (Supplementary Fig. 5e). The virus-induced lysis of salivary gland cells is likely to have affected the shedding of viral particles into the saliva, leading to the observed reduction in feeding and arboviral transmission by the *Ago2*$^{-/-}$ mutants (Fig. 1h, j).

We then employed transmission electron microscopy (TEM) to examine the cell lysis in the midguts of *Ago2*$^{-/-}$ females upon arbovirus infection. Ultrastructural studies showed that the nuclei of the *Ago2*$^{-/-}$ midgut cells displayed an elongated shape, and the nucleolus was not well formed at 4 dpi with MAYV, compared to the WT midgut cells (Fig. 3d), indicating that the cells may undergo nuclear fragmentation and apoptosis. Moreover, clusters of virions were observed in the midgut cells of *Ago2*$^{-/-}$ mutants, whereas the virions were evenly distributed in the WT midguts. At 7 dpi with MAYV, the pattern of infection in the WT midguts was similar to that at 4 dpi, but more virions were observed; in contrast, the midgut epithelial cells of *Ago2*$^{-/-}$ females were completely lysed and devoid of nuclei, and the virus particles appeared to be present throughout the entire cell (Fig. 3e).

In summary, our data demonstrate that *Ago2*$^{-/-}$ mutants have a weakened defense against arbovirus infection and that hyper-infection lyses the mutant mosquitoes' cells and damages their tissues, leading to pathogenesis that compromises their feeding propensity (Fig. 1h) and viability (Fig. 3a).

## Arbovirus hyper-infection stimulates broad-spectrum antiviral immunity in *Ago2*-deficient mosquitoes

To probe to the underlying mechanism governing the infection-induced mortality in *Ago2*$^{-/-}$ mutants, we compared the whole transcriptomes of MAYV-infected *Ago2*$^{-/-}$ and WT females, as well as uninfected *Ago2*$^{-/-}$ and WT females. As compared to the WT mosquitoes, we found that 1,279 genes were upregulated, and 1,128 genes were downregulated in *Ago2*$^{-/-}$ mutants in response to MAYV infection at 4 dpi (Fig. 4a and Supplementary Data 1), and 555 upregulated and 280 downregulated genes in uninfected *Ago2*$^{-/-}$ mutants (Supplementary Data 2).

GO and interacting network analysis of the upregulated genes showed an enrichment of mRNAs corresponding to genes with predicted functions in the categories of immune response, defense, and response to stimuli in *Ago2*$^{-/-}$ mutants at 4 dpi with MAYV when

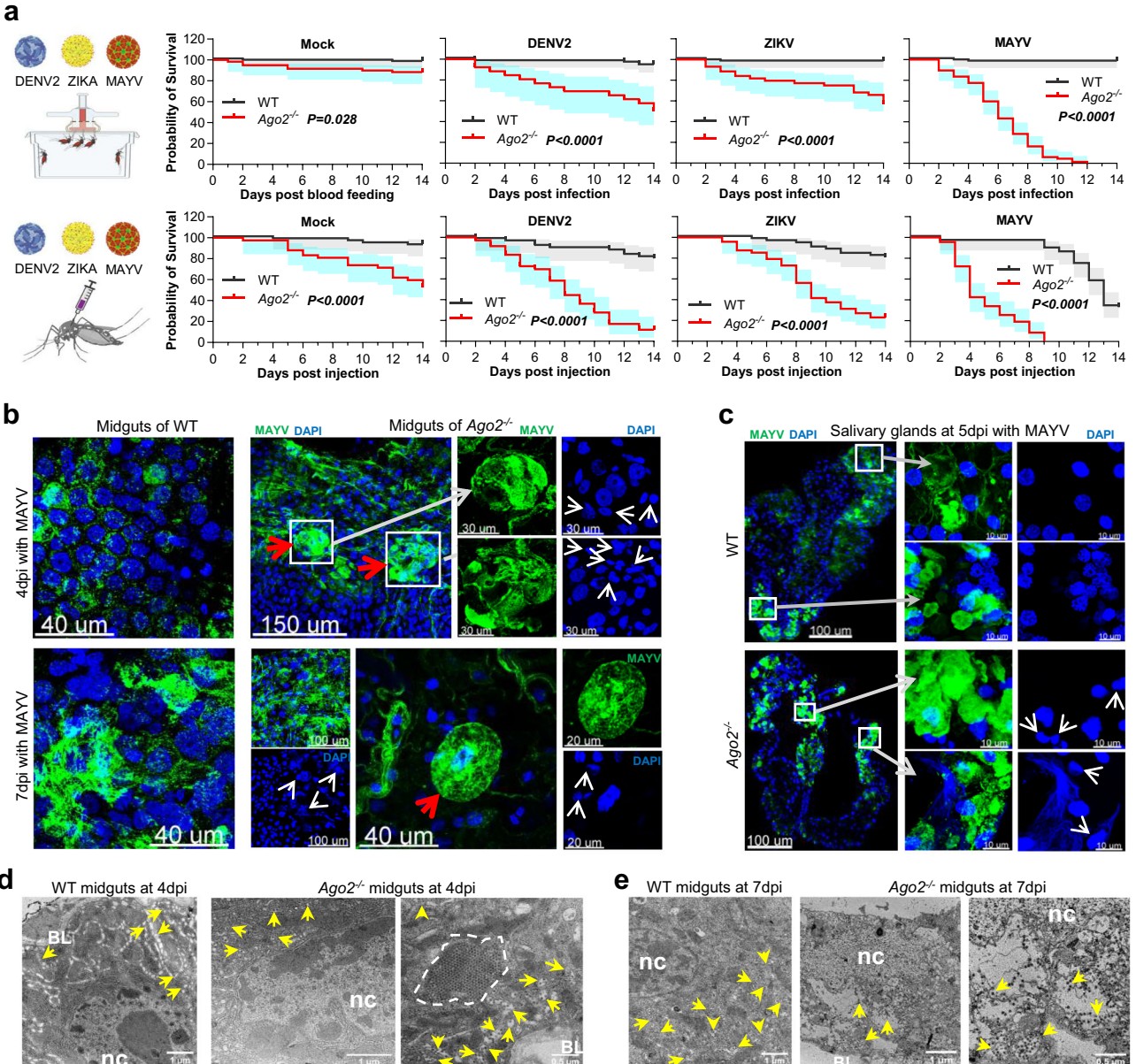

**Fig. 3 | Effect of arbovirus infection on mosquito death and tissue/cell damage in *Ago2⁻/⁻* mutants. a** survival curves of *Ago2⁻/⁻* and wild-type (WT) females that ingested a blood meal containing DENV2, ZIKV, or MAYV or were mock-infected (given C6/36 cell culture medium), or were injected with DENV2, ZIKV, or MAYV or mock-infected medium. *P* values between *Ago2⁻/⁻* mutants and WT mosquitoes were determined using the logrank (Mantel-Cox) test. Each experiment comprised at least three biological replicates. The error bands were indicated by the shaded region. IFA detection with the anti-MAYV monoclonal antibody (green) of MAYV in midguts (**b**) and salivary glands (**c**) of *Ago2⁻/⁻* and WT females at various days post- infection (dpi). Nuclei were stained with DAPI (blue). The zoomed areas are indicated as white boxes and arrows. White arrows indicate abnormal nuclei, and red arrows indicate strong fluorescence with a "ball" shape. Ultrastructural TEM images showing cross-sections of midgut tissue of *Ago2⁻/⁻* mutants and WT mosquitoes at 4 dpi (**d**) and 7 dpi (**e**) with MAYV. Yellow arrows indicate individual virion; white dotted lines outline a cluster of virions. Scale bars are indicated on each image. Abbreviations: BL basal lamina, nc nucleus. The IFA and TEM images in (**b**–**e**) are representative of three biologically independent samples. Source data are provided as a Source Data file.

compared to the WT mosquitoes (Fig. 4b and Supplementary Table 1). A group of immune genes, including caspases, C-type lectin (CTL), and leucine-rich repeat (LRR) family members was significantly enriched in *Ago2⁻/⁻* mutants (Fig. 4c and Supplementary Data 1). Most of the upregulated immune genes such as genes from the CTL, fibrinogen-related protein (FREP), Clip-domain serine protease (CLIP) and pepti-doglycan recognition protein (PGRP) family have immune-defense/ antibacterial functions, but two groups of genes (apoptosis genes and *Dcr2*) have specific antiviral functions, and four groups including genes from the IMD, Toll, JAK/STAT and JNK pathway have both antibacterial and antiviral functions (Fig. 4d). There are four groups of genes with

unknown functions that encode proteins with immunoglobulin (Ig)-like, macroglobulin (Mg) domain, Sushi/SCR/CCP domain, and T/B cell-related features. We also found a large number of putative immune genes with predicate antibacterial function that were significantly upregulated in uninfected *Ago2⁻/⁻* mutants as compared to WT mos-quitoes at 4 days post blood feeding (Supplementary Fig. 6a and Supplementary Data 2), suggesting these groups of genes may not be induced by viral hyper-infection.

At least eight caspase and apoptosis-related genes were upregu-lated, and one inhibitor of apoptosis gene (*IAP4*) was downregulated in *Ago2⁻/⁻* mutants at 4 dpi with MAYV (Supplementary Fig. 6b). The

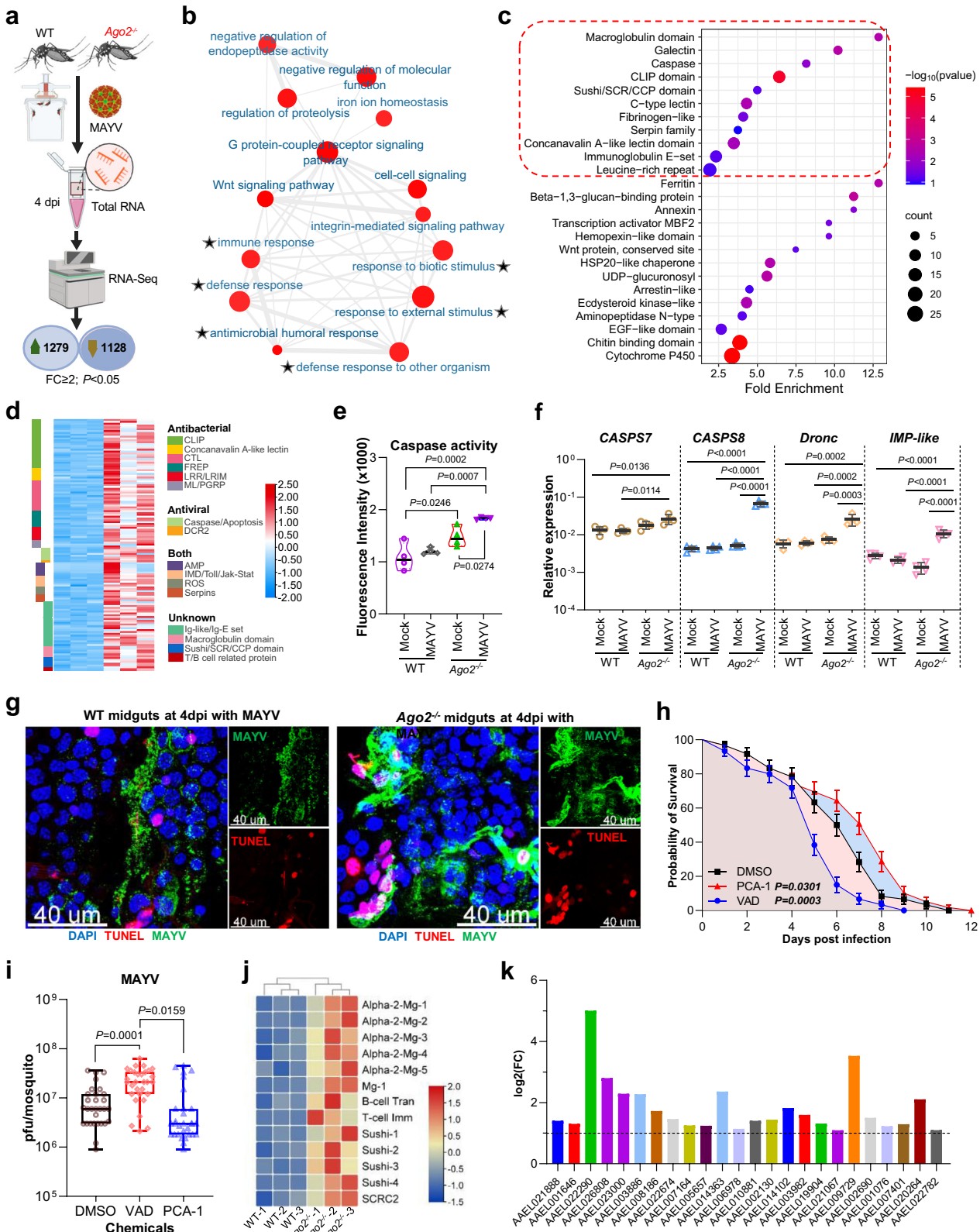

upregulated genes included 6 caspase genes (*CASPS8, CASPS9, CASPS17, CASPS18, CASPS19*, and *Dronc*), *IMP-like* IAP-antagonist Micheob_x-like protein (*IMP-like*, AAEL014196) and nuclear apoptosis-inducing factor 1 (*NAIF1*, AAEL022840) genes. However, only one caspase gene (*CASPS8*) was significantly upregulated in uninfected *Ago2⁻/⁻* mutants as compared to WT mosquitoes (Supplementary Data 2). These data point to an upregulation of the apoptosis

pathway in *Ago2⁻/⁻* mutants in response to MAYV infection. Next, we measured overall caspase activity of *Ago2⁻/⁻* and WT females using a caspase-3 assay kit, and our results showed that caspase activity was increased in the midguts and carcasses of *Ago2⁻/⁻* mutants at 4 dpi with MAYV (Fig. 4e and Supplementary Fig. 6c) and in DENV2-infected *Ago2⁻/⁻* mutants at 14 dpi (Supplementary Fig. 6d) when compared to WT mosquitoes. Transcript abundance analyses with qPCR confirmed

**Fig. 4 | Transcriptomic analysis of antiviral immune signaling in *Ago2*<sup>−/−</sup> mutants upon MAYV infection. a** flowchart illustrating the design and outcome of RNAseq experiments comparing transcriptome changes in *Ago2*<sup>−/−</sup> mutants and WT mosquitoes. DE genes, differentially expressed genes. *n* = 3 biological replicates. **b** interaction network illustrating the upregulated gene families in *Ago2*<sup>−/−</sup> mutants. The light color indicates the small *P*-value (one-sided Fisher's exact test), and the small size of the bubble indicates the few genes in the GO term. The immunity- and defense-related GOs are indicated by asterisks. **c** fold enrichment of the significantly upregulated gene families in *Ago2*<sup>−/−</sup> mutants. The immunity-related gene families are outlined red. **d** heatmap illustrating transcriptome changes in the upregulated immunity genes in *Ago2*<sup>−/−</sup> mutants. **e** detection of caspase activity in midguts of *Ago2*<sup>−/−</sup> and WT females. Data are presented as mean ± SD (*n* = 4). **f** confirmation by qPCR of the upregulated expression of caspase and apoptosis

genes in midguts of *Ago2*<sup>−/−</sup> females. Data are presented as mean ± SD (*n* = 4). **g** TUNEL assay to detect apoptotic cells (red) in midguts of *Ago2*<sup>−/−</sup> and WT females. MAYV antigen is indicated by green, and nuclei are in blue. **h** survival curves for the MAYV-infected WT females that were fed with apoptosis inducer PAC-1 or apoptosis inhibitor Z-VAD-FMK (VAD) or an equal amount of DMSO in 10% sucrose solution. Data are presented as percents ± SE (*n* = 3). **i** virus titer in WT females treated with PAC-1 or VAD at 7 dpi with MAYV. Data were presented as box and whiskers (Min to Max), *n* = 30. **j** upregulated immune genes related to B-cell, T-cell, complement (Suchi), or macroglobulin (Mg) in *Ago2*<sup>−/−</sup> mutants. **k** upregulated immunoglobulin (Ig)-like genes in *Ago2*<sup>−/−</sup> mutants. *P* values were determined by one-way ANOVA (**e** and **f**), an unpaired two-sided Mann-Whitney test (**i**), or a log-rank (Mantel-Cox) test (**h**). Source data are provided as a Source Data file.

that the expression of caspase and *IMP-like* genes was significantly induced in the midguts and carcasses of *Ago2*<sup>−/−</sup> females at 4 dpi with MAYV (Fig. 4f and Supplementary Fig. 6e) and in DENV2-infected *Ago2*<sup>−/−</sup> females at 14 dpi (Supplementary Fig. 6f). As compared to the WT mosquitoes, the midguts of the *Ago2*<sup>−/−</sup> mutants had more apoptotic cells (by TUNEL staining) (Fig. 4g and Supplementary Fig. 6g). The *Ago2*<sup>−/−</sup> females exhibited a lower and higher probability of survival after MAYV infection (Fig. 4h) when treated with an apoptosis inhibitor (Z-VAD-FMK) or inducer (PAC-1), respectively, and the virus titer was significantly increased or reduced in the inhibitor- or inducer-treated females (Fig. 4i). These results together demonstrate that apoptosis is stimulated in *Ago2*<sup>−/−</sup> mutants upon arbovirus infection, and these mutants may use an alternative antiviral mechanism against arboviral infection.

Another group of upregulated immune genes have a predicted function related to the mammalian immune system, including B/T cell-mediated humoral/cellular immunity and complement. These genes were highly induced in *Ago2*<sup>−/−</sup> mutants at 4 dpi with MAYV (Fig. 4j–k).

Overall, this stimulation of broad-spectrum antiviral genes in *Ago2*<sup>−/−</sup> mutants upon arbovirus infection suggests that mosquitoes may engage other antiviral defense mechanisms to protect themselves from viral infection when the major antiviral siRNA pathway has been disrupted.

## *Ago2* disruption impairs DNA repair and replication and reduces histone abundance

GO and interacting network analyses of the downregulated genes showed that genes with functions related to DNA replication, repair, and metabolism were significantly downregulated in *Ago2*<sup>−/−</sup> mutants at 4 dpi with MAYV (Fig. 5a and Supplementary Table 2), indicating that the DNA replication and repair mechanisms are significantly impaired in *Ago2*<sup>−/−</sup> mutants. Then, we compared the expression of genes with predicated function in DNA damage response (DDR) that controls lesion detection and DNA repair[27] between *Ago2*<sup>−/−</sup> mutants and WT mosquitoes. We found that most of the DDR genes that are involved in base excision repair (BER), double-strand break (DSB) repair, DNA mismatch repair (DMR) and nucleotide excision repair (NER) were downregulated in *Ago2*<sup>−/−</sup> mutants in response to MAYV infection at 4 dpi (Fig. 5b, c and Supplementary Data 1), as compared to the WT mosquitoes. DMR and NER are the largest groups of the downregulated DNA repair genes. Most DDR genes were also downregulated and 8 of them were significantly downregulated in uninfected *Ago2*<sup>−/−</sup> mutants as compared to WT mosquitoes at 4 days post blood feeding (Supplementary Fig. 7a and Supplementary Data 2), suggesting *Ago2* disruption may have a direct impact on DNA repair. The downregulation of DDR genes was further confirmed by qPCR in *Ago2*<sup>−/−</sup> females with or without MAYV infection at 4 dpi (Fig. 5d), and with or without DENV2 infection at 14 dpi (Supplementary Fig. 7b), as compared to the WT controls.

Taken together with our observation from the midgut TEM and TUNEL studies that the nuclei of the midgut cells in the mutant

mosquitoes had an abnormal shape (Fig. 3d, e) and underwent apoptosis and DNA fragmentation (Fig. 4g and Supplementary Fig. 6g), our results here suggest that hyper-infection of arboviruses induces DNA damage in *Ago2*<sup>−/−</sup> mutants, and *Ago2* disruption impairs DNA repair mechanisms.

Several histone genes were among those downregulated in *Ago2*<sup>−/−</sup> mutants (Fig. 5e), and the expression of at least 20 histone genes was significantly reduced by more than 2-fold in *Ago2*<sup>−/−</sup> mutants at 4 dpi with MAYV (Fig. 5f). Next, we compared the mRNA abundance of histones and histone-modifying enzyme genes between *Ago2*<sup>−/−</sup> mutants and WT mosquitoes. Our results showed that the majority of these genes were downregulated in MAYV infected (Fig. 5g and Supplementary Data 1) and uninfected *Ago2*<sup>−/−</sup> mutants (Supplementary Fig. 7c, d and Supplementary Data 2), and a reduction of up to 36.6% at 4 dpi with MAYV (Fig. 5h) and 24.59% without infection (Supplementary Fig. 7e), suggesting that impact of *Ago2* disruption on histone abundance is independent of viral infection and downregulation of histones may be responsible for the impaired DNA replication and repair mechanisms in *Ago2*<sup>−/−</sup> mutants. Next, we used qPCR to examine the expression of three histone core genes (*His2A*, *His3* and *His4*) that were significantly downregulated in the transcriptome of MAYV infected *Ago2*<sup>−/−</sup> mutants. Our results showed that the mRNA abundance of these histone genes was significantly decreased in the midguts and carcasses of *Ago2*<sup>−/−</sup> females, with or without MAYV infection (Fig. 5i and Supplementary Fig. 7f), and with or without DENV2 infection (Supplementary Fig. 7g), when compared to the corresponding WT mosquitoes, indicating that *Ago2* disruption affects the expression of histones, independent of arbovirus infection.

MAYV infection reduced the expression of histone genes in carcasses and midguts of the WT mosquitoes when compared to non-infected mosquitoes (Fig. 5i and Supplementary Fig. 7g), suggesting that MAYV infection affects histone expression. To better understand the interaction between histones and arbovirus infection, we silenced three histone genes (*His2A*, *His3*, and *His4*) by injecting dsRNAs into the WT mosquitoes prior to virus infection. When compared to the GFP control, silencing of each histone gene significantly reduced the MAYV titer (Fig. 5j). When the three histone genes were co-silenced, a significant 2.75-fold (*P* < 0.0001) reduction in the MAYV titer was observed (Fig. 5j). Furthermore, when the WT mosquitoes were treated with a histone deacetylase (HDAC) inhibitor (Scriptaid), which inhibits HDAC to remove acetyl groups from the lysine residues of histone and HDAC inhibitor treatment increases genome-wide histone acetylation[28], they exhibited a significantly higher MAYV titer (Fig. 5k) and lower survival rate (Fig. 5l) than did those mosquitoes treated with a DMSO control. These data indicate that silencing of histones results in a decreased virus titer.

To address whether histone downregulation leads to the impaired DDR in *Ago2*<sup>−/−</sup> mutants, we co-silenced three histone genes (*His2A*, *His3*, and *His4*) in WT mosquitoes and then assayed mRNA abundance of DNA repair genes. Our results show that silencing of histone genes

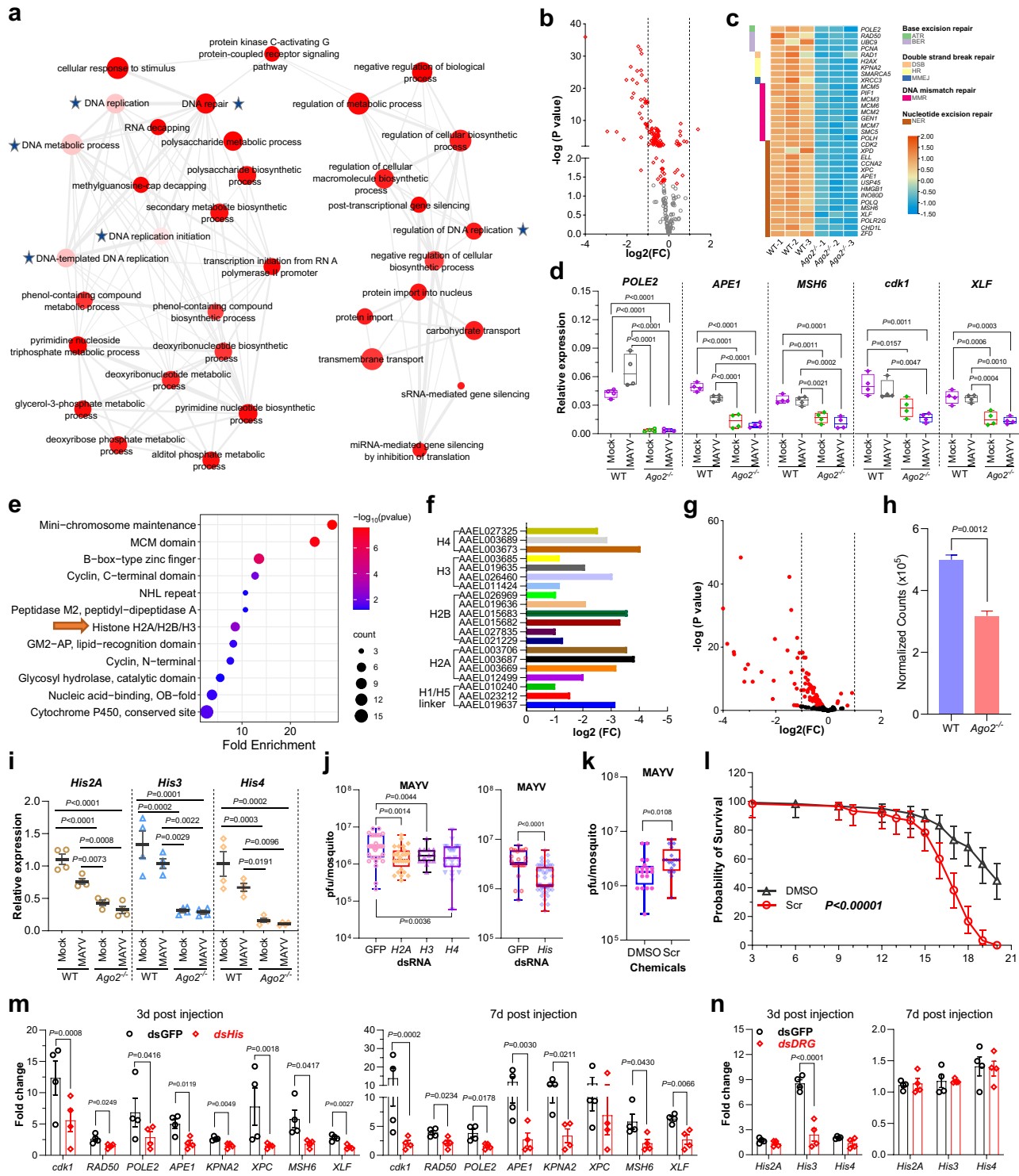

significantly affects mRNA abundance of all the DNA repair genes at 3 days post dsRNA injection and 7 of 8 DNA repair genes at 7 days post dsRNA injection (Fig. 5m). On the other hand, co-silencing of DNA repair genes (*cdk1*, *RAD50*, *APE1* and *KPNA2*) reduces the mRNA abundance of *His3* at 3 days post dsRNA injection but does not affect mRNA abundance of all three histone genes at 7 days post dsRNA injection even though silencing of DNA repair genes persists at that time point (Fig. 5n and Supplementary Fig. 8a). Then, we silenced individual DNA repair gene and investigated its impact on histone gene mRNA abundance. We found that silencing of *cdk1* had the most effect on histone gene mRNA abundance at 3 days post dsRNA injection

(Supplementary Fig. 8b). At 7 days post dsRNA injection, gene silencing was only persistent in *RAD50* dsRNA-injected mosquitoes, but its silencing increased expression of *His2A* and *His3* (Supplementary Fig. 8c). Hence, the effect of DNA repair gene silencing on histone gene expression is inconclusive. Taken together, our data support the hypothesis that histone gene downregulation is the major reason for the observed defects in DNA replication and repair mechanisms in *Ago2*[−/−] mutants.

In summary, *Ago2* disruption reduces the production of histones and compromises the DNA replication and repair mechanism in the arboviruses-infected mosquitoes. The downregulation of histone

**Fig. 5 | Transcriptomic analysis of DNA repair and histone genes in *Ago2*⁻/⁻ mutants upon MAYV infection and effect of histone reduction on virus infection and mosquito mortality. a** interaction network illustrating downregulated gene families in *Ago2*⁻/⁻ mutants. The light color indicates small *P*-value (one-sided Fisher's exact test), and the small size indicates few genes in the GO term. GOs related to DNA repair and replication are indicated by asterisks. **b** volcano plot illustrating fold changes in DNA repair genes (*DRG*s) in *Ago2*⁻/⁻ mutants. **c** heatmap illustrating transcriptome changes in upregulated *DRG*s in *Ago2*⁻/⁻ mutants. **d** qPCR detecting expression of downregulated *DRG*s in *Ago2*⁻/⁻ carcasses. **e** fold enrichment of downregulated gene families in *Ago2*⁻/⁻ mutants. **f** downregulated histone genes in MAYV infected *Ago2*⁻/⁻ mutants. **g** volcano plot illustrating fold changes of histones-related genes in *Ago2*⁻/⁻ mutants. **h** overall histone level in MAYV infected *Ago2*⁻/⁻ and WT mosquitoes. **i** qPCR detecting expression of histone genes in carcasses of *Ago2*⁻/⁻ and WT mosquitoes at 4 days post-infection (*n* = 4). **j** virus titer in histone dsRNA-injected WT females at 7 dpi as determined by plaque assay. Left: *n* = 35 (dsGFP), *n* = 28 (ds*His2A*), *n* = 15 (ds*His3*) and *n* = 26 (ds*His4*). Righ: *n* = 29 (dsGFP) and *n* = 51 (ds*His*). **k** virus titer in WT mosquitoes treated with histone deacetylase inhibitor Scriptaid (Scr), *n* = 20. **l** survival curves of MAYV-infected WT females that were fed with Scriptaid or DMSO in 10% sucrose solution. Data are represented as percents ± SE (*n* = 3). qPCR detecting *DRG* expression (**m**) and histone expression (**n**) in WT females that were injected with ds*His* (mixture of dsRNAs of *His2A*, *His3* and *His4*) or ds*DRG* (mixture of dsRNAs of *cdk1*, *RAD50*, *APE1* and *KPNA2*) at 3- and 7-days post injection, respectively. *n* = 4. *P* values were determined by a one-way ANOVA (**d**, **i**), an unpaired two-sided *t*-test (**h**), two-way ANOVA with multiple comparisons (**m** and **n**), a logrank (Mantel-Cox) test (**l**), or an unpaired two-sided Mann-Whitney test (**j**, **k**). Data are presented as mean ± SEM (**h**, **i**, **m** and **n**), and box and whiskers (Min to Max) (**d** and **j**) from 4 replicates. Source data are provided as a Source Data file.

genes may play a protective role during arbovirus infection, likely through impairing the replication mechanism of the arboviruses.

## *Ago2* disruption represses autophagy by altering histone abundance

We found that Ref(2)P (polyubiquitin binding protein p62), which serves as a marker of autophagic flux in *Drosophila*[29,30], was significantly upregulated, and two autophagy-related genes (*Atg*) were downregulated in the transcriptomes of *Ago2*⁻/⁻ mutants upon MAYV infection (Fig. 6a). Most autophagy-related genes were also downregulated in uninfected *Ago2*⁻/⁻ mutants and one of them (*Atg10*) was strongly downregulated (Supplementary Fig. 9a and Supplementary Data 2). Then, we used qPCR to examine the expression of a panel of *Atg* (*Atg5*, *Atg7*, *Atg10*, and *Atg12*), *p62* and *P53*, a transcription factor that transactivates a panel of *Atg* genes in *Drosophila*[31], between *Ago2*⁻/⁻ mutants and WT mosquitoes in response to infection with different arboviruses. Our results confirmed the downregulation of *Atg* and *P53* in *Ago2*⁻/⁻ mutants with or without MAYV infection at 4 dpi (Fig. 6b), and with or without DENV2 infection at 14 dpi (Supplementary Fig. 9b), as compared to the WT controls. We also observed downregulation of *p62* in uninfected *Ago2*⁻/⁻ mutants as compared to the WT mosquitoes and MAYV infection increased *p62* expression in *Ago2*⁻/⁻ mutants (Fig. 6b). The protein levels of ATG8 and phosphatidylethanolamine (PE) of ATG8 (ATG8-PE), and P53 were also reduced in *Ago2*⁻/⁻ mutants as compared to the WT controls (Fig. 6c and Supplementary Fig. 9c–f). These data suggest a less active autophagy pathway in *Ago2*⁻/⁻ mutants.

To further confirm the repressed autophagy in *Ago2*⁻/⁻ mutants, we used Lysotracker Red (LTR), a vital dye used for the detection of acidic organelles (including autolysosomes), to stain the midguts from *Ago2*⁻/⁻ and WT females at 4 dpi with MAYV. As compared to the WT mosquitoes, in which numerous LTR-positive dots accumulated in the midgut epithelium, the midguts of *Ago2*⁻/⁻ females showed a strongly reduced accumulation of LTR-positive cells (Fig. 6d) and LTR integrated density (Fig. 6e) upon MAYV infection. TEM was further used to document autophagy defects in the midgut cells of *Ago2*⁻/⁻ females. At 4 dpi with MAYV, many autophagosomes and autolysosomes were observed in the midgut cells of the WT mosquitoes, whereas the midgut cells of *Ago2*⁻/⁻ mutants exhibited a strong reduction in these autophagic structures, as well as numerous areas representing black cell debris/aggregates (Fig. 6f). A large number of autophagosomes and autolysosomes were also present in the midgut cells of the WT mosquitoes at 7 dpi with MAYV (Fig. 6g). Because of the strong cell lysis induced in *Ago2*⁻/⁻ mutants at 7 dpi with MAYV, we saw no autophagosomes or autolysosomes, but cell debris and black aggregates were present in both the cytoplasm and the nuclei (Fig. 6g). Taken together, these results demonstrate that *Ago2* disruption severely impairs the induction of autophagy upon arbovirus infection.

To confirm the role of autophagy in inhibiting arbovirus infection in mosquitoes, we silenced *P53* and *p62* in the WT mosquitoes prior to infection and determined the virus titer in the silenced mosquitoes. The results showed that the MAYV titer was significantly elevated in the *P53*-silenced mosquitoes and suppressed in the *p62*-silenced mosquitoes (Fig. 6h). When *Ago2*⁻/⁻ mutants were treated with the autophagy inhibitor 3-MA or an inducer, rapamycin, they exhibited a significantly lower or higher probability of survival, respectively, upon MAYV infection (Fig. 6i). These data indicate that autophagy plays a role in preventing mosquito death during arbovirus infection by inhibiting viral infection.

The regulation of autophagy is linked to histone modification and altered histone abundance in mammals[32,33]. To test the effect of histone loss on autophagy in mosquitoes, we silenced three histone genes (*His2A*, *His3*, and *His4*) by injecting a mixture of dsRNAs for these genes into the WT mosquitoes and monitoring the *Atg* expression by qPCR in the silenced mosquitoes. As compared to the GFP control, silencing the histone genes significantly reduced the expression of a panel of autophagy genes (*Atg5*, *Atg7*, *Atg10*, and *Atg12*) in both sugar-fed and blood-fed mosquitoes (Fig. 6j), and these results were correlated with the silencing efficiency of the histone genes (Supplementary Fig. 9g). As compared to the sugar-fed mosquitoes, the blood-fed mosquitoes exhibited a significantly higher expression of *Atg5* and *Atg10* in the GFP control mosquitoes, but not in the *histone*-silenced mosquitoes.

In summary, *Ago2* disruption significantly repressed the autophagy pathway in mosquitoes, most likely by affecting histone production, and this reduced autophagy promoted mosquito death by hampering the removal of damaged cell aggregates in the arbovirus-hyper-infected mosquitoes.

## A proposed model of arbovirus-induced mosquito death mechanism

We propose a working model to summarize our findings and to explain an arbovirus-induced mosquito death mechanism in *Ago2*⁻/⁻ mutants (Fig. 7): (1) *Ago2* disruption does not affect the biogenesis of small RNA but impairs the arboviral RNA degradation through a defective siRNA pathway in *Ae. aegypti*; (2) *Ago2* disruption leads to arbovirus hyper-infection of mosquitoes, which is accompanied by cell lysis and results in mortality; (3) *Ago2* disruption influences the production of histones, which represses the autophagy and DNA repair mechanism in arboviruses-hyper-infected mosquitoes. (4) hyper-infection by arboviruses and defect in DNA repair along with cytoplasmic degradation promote mosquito death.

## Discussion

Here we show that the *Ae. aegypti* Ago2 is essential for controlling arbovirus infection, and thereby an important regulator of mosquito disease transmission. We describe a mechanism to account for the lethality caused by arbovirus infection in siRNA pathway-defective mosquitoes (*Ago2*⁻/⁻ mutants), in which hyper-infection leads to lysis of mosquito cells and damage to tissues, while at the same time the DNA repair/replication mechanism and elimination of damaged and

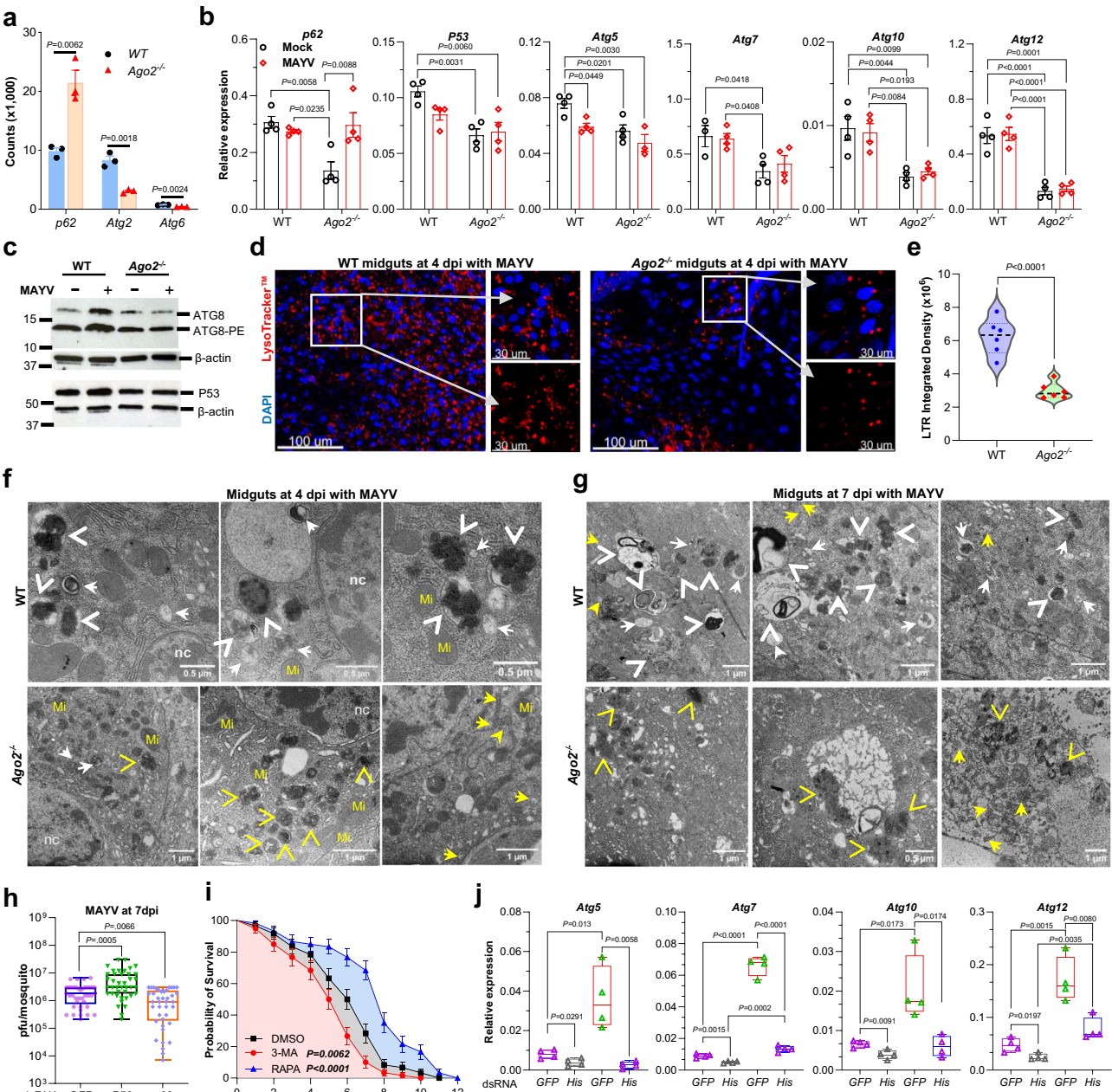

**Fig. 6 | Effect of *Ago2* disruption and histone loss on autophagy in mosquitoes upon MAYV infection. a** upregulated *p62* and downregulated autophagy-related genes (*Atg*) in the transcriptome of *Ago2*[−/−] mutants. *n* = 3. **b** expression of *p62*, *P53* and *Atg* genes in the carcasses of *Ago2*[−/−] and WT females at 4 days post-infection (dpi) with MAYV, as detected by qPCR. *n* = 4 biological replicates. **c** western blot showing protein levels of ATG8, ATG8-phosphatidylethanolamine (ATG8-PE) and P53 in *Ago2*[−/−] and WT females at 4 dpi with MAYV; β-actin was used as a loading control. Western blots were repeated at least three times, and representative results are shown. Lysotracker Red (LTR) staining (**d**) showing the impairment of autophagy and LTR integrated density (**e**) in midguts of *Ago2*[−/−] females at 4 dpi with MAYV. Nuclei were stained with DAPI (blue). *n* = 6 biologically independent samples. Autolysosomes (arrowheads) and autophagosomes (white arrows) were detected by TEM in the midguts of *Ago2*[−/−] and WT females at 4 dpi (**f**) and 7 dpi (**g**) with MAYV. Yellow arrows indicate virions, and yellow arrowheads indicate cell debris. Abbreviations: Mi mitochondria, nc nucleus. *n* = 3 biologically independent samples. **h** virus titer in *P53*- or *p62*- dsRNA-injected WT females at 10 dpi with MAYV as determined by plaque assay. dsRNA for GFP was injected as a control. Data were presented as box and whiskers (Min to Max), *n* = 41 (dsGFP), *n* = 40 (ds*P53* and ds*p62*). **i** survival curves of the MAYV-infected WT females that were fed with autophagy inhibitor 3-MA or inducer rapamycin (RAPA) or an equal amount of DMSO in 10% sucrose solution. Data were presented as percents ± SE (*n* = 3). **j** qPCR detecting the expression of a panel of *Atg* genes in histone dsRNA-injected WT mosquitoes at 3 days post-injection and 4 days post-blood-feeding. *n* = 4 biological replicates. *P* values were determined by an unpaired two-sided *t*-test (**a**, **e** and **j**), one-way ANOVA for (**b**), an unpaired two-sided Mann-Whitney test for (**h**), and a logrank (Mantel-Cox) test (**i**). Data are represented as mean ± SEM (**a** and **b**). Source data are provided as a Source Data file.

dysfunctional cytoplasmic materials are severely impaired. Further analysis revealed that the reduced abundance of histones in *Ago2*[−/−] mutants may compromise their DNA repair mechanism and autophagy. Arbovirus-induced mortality was also observed in *Dicer2*[−/−] mutant mosquitoes that also have an impaired siRNA pathway, but

mortality of *Dicer2*[−/−] mutants differs between infections with different arboviruses; e.g. mortality upon chikungunya virus (CHIKV) infection vs no significant mortality upon DENV3 infection[34] and high mortality upon Sindbis virus (SINV) infection vs low mortality upon yellow fever virus (YFV) infection[35]. It appears that infections by alphaviruses

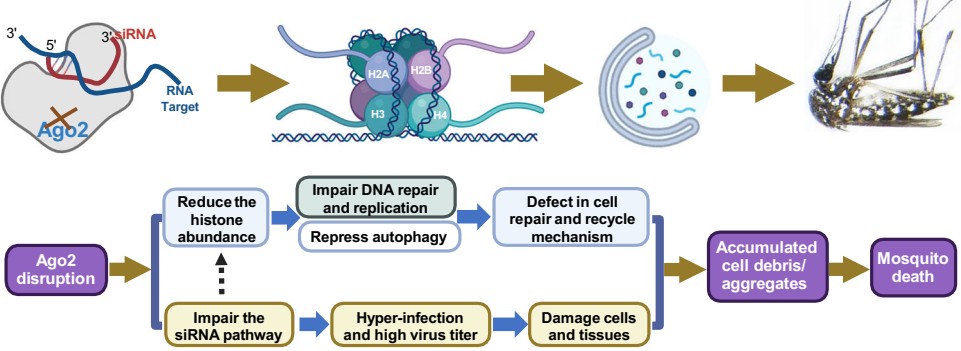

**Fig. 7 | A proposed model summarizing arboviruses-induced death in *Ago2*-deficient *Ae. aegytpi*.**

(CHIKV, SINV and MAYV) induces a higher mortality in siRNA-deficient mosquitoes as compared to flaviviruses (DENV2, DENV3, ZIKV and YLV). It is impossible to quantitatively compare arbovirus-induced mortality in *Ago2*[−/−] and *Dicer2*[−/−] mutants because of differences in arboviruses, viral titers, infection methods (systemic injection vs blood feeding) used in these independent studies.

In the baculovirus-insect system, infection alters insect physiology and behavior[36] and usually causes lethality in the larval stages of some lepidopterans, mosquitoes, and sawflies[37]. For example, in the fall armyworm (*Spodoptera frugiperda*), larval infection with baculovirus blocks molting as a result of the production of a viral enzyme, ecdysteroid UDP-glucosyltransferase (EGT), which prevents larval molting by inactivating ecdysone[38]. In contrast to the baculoviruses-insect system, infection with arboviruses in mosquitoes is nonpathogenic, preventing vaccinated strains of arboviruses from potentially being used as biological insecticides for mosquito control, as baculoviruses are. Our results here show that impairment of the siRNA pathway renders mosquitoes hypersensitive to infection by arboviruses and converts a nonpathogenic infection with arboviruses in mosquitoes into a lethal infection. Importantly, the vector competence (transmission potential) of arboviruses in *Ago2*[−/−] mutants was significantly lower than that of WT mosquitoes despite the fact that the mutant mosquitoes had a higher titer of arboviruses, which suggests that this type of siRNA pathway disruption could be explored for the development of novel arbovirus transmission-blocking strategies.

Our results demonstrate that the siRNA pathway is an essential determinant of persistent infection by arboviruses in mosquitoes. Unlike vertebrates with a well-developed immune system, insects lack adaptive or acquired immune responses but have constitutive and induced systemic and local innate immune defenses against pathogen infection[39,40]. In *Drosophila*, the siRNA pathway is considered a constitutive defense system against virus infection, whereas the other innate immune pathways are inducible by infection with various pathogens[20]. Here we also show that when the siRNA pathway is impaired, other conserved immune signaling cascades, including the Toll and apoptosis pathways, are induced upon virus infection or other stresses and injuries and may play a role in defending against viral infection. A siRNA pathway deficiency appears to be compensated by complementary antiviral response comprising diverse defense systems that, however, will be inadequate to suppress virus infection to a tolerable level. In addition to induction of classical innate immune signaling, hyper-infection of arboviruses also stimulates induction of immune genes with macroglobulin and Sushi/SCR/CCP domains, and these genes have predicted functions related to mammalian immune signaling, including B/T cell-mediated humoral/cellular immunity and complement, but their antiviral role has not been explored in mosquitoes or other insects.

We report that *Ago2* disruption severely impairs the degradation of viral dsRNA, resulting in high infection intensities in *Ago2*[−/−] mutants,

which demonstrates the essential role of *Ago2* in the mosquito siRNA pathway. Meanwhile, the production of viral small interfering RNAs (vsiRNAs) is increased in *Ago2*[−/−] mutants, indicating that *Ago2* disruption does not affect the endonuclease activity of Dcr2 to process viral dsRNA into 21-nt vsiRNA as shown in *Drosophila*[41,42]. Despite the high amount of vsiRNA in *Ago2*[−/−] mutants, their failure to form RNA-induced silencing complexes (RISCs) because of a lack of Ago2 prevents the RISC-mediated degradation of the viral dsRNA. In our previous study, we have shown that overexpression of *Dcr2* in mosquito midguts significantly reduces the production of vsiRNA as a result of the low viral titer in transgenic mosquitoes[43], indicating that the production of vsiRNA is dependent on both the arbovirus titer and Dcr2 activity. Therefore, Dcr2 activity appears to be independent of Ago2, and Dcr2 functions upstream of the siRNA production. According to our findings, the abundance of mosquito 21nt-siRNAs is also increased in *Ago2*[−/−] mutants as compared to WT mosquitoes, suggesting that *Ago2* disruption may adversely affect the degradation of endogenous mosquito dsRNAs. Taken together, our results show that Ago2 functions downstream of the siRNA pathway in mosquitoes and plays key roles in degrading endogenous (mosquito) and exogenous (arbovirus) dsRNAs.

DNA repair and replication mechanisms and histone abundance are significantly affected by arbovirus infection in *Ago2*[−/−] mutants. Considering the role of histones in mediating DNA repair and replication[44–46], the impaired DNA repair and replication in *Ago2*[−/−] mutants may be a result of the decreased abundance of histones. Histone loss in *Ago2*[−/−] mutants might be caused by hyper-infection of arboviruses and the resulting induction of DNA damage, as has been reported in in yeast[47]. However, a reduced expression of histone genes when compared to WT mosquitoes was also observed in *Ago2*[−/−] mutants in the absence of arbovirus infection. These data indicate that *Ago2* disruption may have a direct impact on histone production in mosquitoes, suggesting that Ago2 may play a role in modifying nucleolus organization through affecting histone production and modification, as has been reported in *Drosophila* and mammals, in which Ago2 is involved in chromatin topology[19] and modification[48,49], and chromatin changes can be modulated by histone variants and modification[50–52].

Autophagy is a highly conserved process in eukaryotic cells for maintaining cellular and tissue homeostasis by degrading damaged organelles and unfolded proteins, and this pathway plays a pivotal role in cells by allowing them to recycle or clear cytoplasmic organelles or cytosolic aggregates under conditions of starvation or other types of cellular stress[53,54]. Autophagy can also be stimulated by infection with intracellular pathogens, including viruses, and it plays a protective role in cells during pathogen infection in *Drosophila*[55–57]. In mosquito cells, infection with DENV2 and CHIKV induces autophagy[58], and treatment with the autophagy inducer rapamycin can activate the autophagy pathway to defend the mosquitoes against DENV infection[59]. Our

results now show that *Ago2* disruption in adult mosquitoes significantly represses autophagy and enhances mosquito death most likely because the cells fail to clear the damaged organelles and other cytoplasmic components that are produced by hyper-infection with arboviruses. The repressed autophagy may result from the impaired histone production in *Ago2*[−/−] mutants, which is consistent with the role of histones in controlling autophagic flux in mammals, where reduced histone modification and histone variants are associated with dysfunction in autophagic processes and sustained autophagy[32,33,60]. Taken together, our data strongly support the conclusion that autophagy plays a protective role during arbovirus infection in *Ae. aegypti* by degrading damaged cell aggregates and inhibiting viral infection, and *Ago2* disruption weakens this protective role by affecting histone production and rendering the mosquitoes vulnerable to viral infection.

## Methods

### Ethics statement

This study was carried out in accordance with the recommendations in the Guide for the Care and Use of Laboratory Animals of the National Institutes of Health, the Animal Care and Use Committee (ACUC) of the Johns Hopkins University, and the institutional Ethics Committee (permit number: M006H300). The IACUC committee approved the protocol. Mice were only used for mosquito rearing. Commercial, anonymous human blood was used for virus infection assays in mosquitoes, and informed consent was therefore not applicable.

### Mosquitoes

The germline *Cas9*-expressed *Ae. aegypti* line (*Cas9*)[61] was shared by Dr. Omar Akbari. All the mosquitoes used in these experiments, including wildtype (Liverpool strain), *Cas9*, and *Ago2* knockout mutant lines were reared at 27 °C under 85% relative humidity with a 12 h light/12 h dark cycle in a walk-in chamber in the Johns Hopkins Malaria Research Institute. Mice (6 to 12 weeks old) were used for blood feeding and colony maintenance and reared at 27 °C under 50% relative humidity with a 12 h light/12 h dark cycle.

### Generation of *Ago2* knockout mutant lines using CRISPR/Cas9

Guide RNA (gRNA) design and synthesis have been described[62]. In brief, the *Ago2* gRNAs were designed using the chopchop CRISPR/Cas9 (https://chopchop.cbu.uib.no/). Each of two gRNAs separated by around 300-bp were designed to target the same *Ago2* exon sequence to achieve a large mutation. The double-stranded DNA templates for each gRNA were generated by template-free PCR, and the purified PCR products were in vitro-transcribed using a HiScribe™ T7 Quick High Yield RNA Synthesis Kit (NEB). Following transcription, the gRNAs were purified using a RNeasy kit (Qiagen), eluted with nuclease-free water, and stored at −80 °C.

For embryo injection, each gRNA was diluted to 100 ng/μl with E-Toxate™ Water (Sigma) and injected into *Cas9* embryos according to our previously established protocol[62]. A leg PCR was used to identify mutations in the surviving G0 adults using a pair of primers ~100 bp outside of each gRNA (Supplementary Table 3). The mutation was identified by an extra band on a 2% agarose gel when compared to the *Cas9* line. The mutated G0 was crossed back to *Cas9* line, and mutation of their offspring (G1) was identified by leg PCR. Each mutated G1 male or female was individually outcrossed with the corresponding *Cas9* female or male to establish various mutant lines. After another four generations, the heterozygotes were in-crossed. The homozygous mutant lines were identified by leg PCR and confirmed by DNA Sanger sequencing. The homozygous mutant females were also crossed to the WT males for another six generations to establish the WT-genetic background of the *Ago2* mutant lines.

### Virus infection, transmission, and titration

Virus infection of mosquitoes and plaque assays were performed as previously described[63]: MAYV (Iquitos, IQT), ZIKV (Cambodia, FSS13025) or DENV serotype 2 (New Guinea C strain, DENV2) was propagated in C6/36 cells. The virus-infected cell culture medium was harvested and mixed with an equal volume of commercial human blood supplemented with 10% human serum containing 10 mM ATP. One-week-old females were starved overnight and transferred into a small cup, then fed for 30 min with the blood-virus mixture at 37 °C on a single glass artificial feeder. Fully engorged females were selected and maintained on 10% sterile sucrose solution. Each experiment had at least two biological replicates, and each replicate included at least 20 individual females.

Viral transmission in mosquitoes was assayed using an artificial glass feeder according to our previously established method[62]. In brief, a cup of eight mosquitoes were fed on a sterilized glass feeder with 200 μl of feeding solution (equal volumes of human serum and DMEM together with 10 mM ATP [final concentration]) for 30 min at 5 dpi with MAYV. Forced salivation was used to collect saliva from individual females[62], and more than 20 mosquitoes were assayed for each strain. The virus titers in the feeding solution and individual saliva were determined by plaque assay. At least two biological replicates were performed for each experiment, and the data were pooled from different replicates and used for generating graphs.

Plaque assays were used to determine the virus titer as previously described[21,63]. In brief, mosquitoes were homogenized in 300 μl of DMEM with a Bullet Blender (Next Advance Inc., Averill Park, NY, USA) with glass beads. They were then centrifuged at 10,000 g for 3 min, and 20 μl of the filtered supernatants were used for plaque assays in 24-well plates, using BHK 21 cells for DENV2 and ZIKV and Vero cells for MAYV. The plates were incubated for 2 days (MAYV), 4 days (ZIKV), or 5 days (DENV2), and then the plaques were visualized by staining with 1% crystal violet and counted under a microscope. At least two biological replicates were performed, and the data were pooled from different replicates and used for generating graphs.

Differences in virus titer and prevalence of infection between different samples were compared by using a non-parametric two-sided Mann-Whitney U-test and Fisher's exact test, respectively.

### Gene silencing assays and qRT-PCR

dsRNA was synthesized using a HiScribe T7 in vitro transcription kit (New England Biolabs) with gene-specific primers (Supplementary Table 3) and purified using a RNeasy kit (Qiagen). Four- or five-day-old females were cold-anesthetized and injected intrathoracically by using the Nanoject II injection system (Drummond Scientific) with about 0.5 μg of dsRNA per mosquito. Two days after dsRNA injection, the females were given a blood-virus mixed meal for infection. dsGFP-injected mosquitoes were used as a control. The gene silencing efficiency was monitored by qPCR. At least three replicates were performed. The embryos of WT and *Ago2* knockout mutants were injected with 1 μg/μl of dsRNA, and larvae hatched from dsRNA-injected eggs were photographed. dsGFP-injected eggs were used as a control. At least 100 eggs were injected from each condition, and two biological replicates were performed.

Total RNA was extracted from mosquitoes or tissues with Trizol reagent (Invitrogen, Carlsbad, CA, USA). First-strand cDNA was synthesized using a QuantiTect Reverse Transcription Kit (Qiagen, Hilden, Germany). qPCR amplification was performed with gene-specific primers (Supplementary Table 3) on a StepOnePlus Real-Time PCR System (Applied Biosystems, Warrington, UK). The relative abundance of the gene transcripts was normalized and calculated by comparison to the ribosomal protein *S7* gene as an endogenous reference using the $2^{-\Delta CT}$ or $2^{-\Delta\Delta CT}$ method. Each sample had at least three independent biological replicates. The significance (*P*-values) between samples was

determined by using an unpaired two-sided *t*-test with GraphPad Prism.

## Western blotting, immunofluorescence (IFA), and TUNEL assays

Total proteins were extracted from mosquitoes with RIPA extraction buffer (Thermo Scientific) supplemented with a 1% cOmplete™ Protease Inhibitor Cocktail (Roche Applied Science). A total of 20 μg of total protein were used in blotting as described previously[62]. Primary antibodies for western blotting were as follows: *Ae. aegypti* anti-ATG8 polyclonal antibody[64] (a gift from Dr. Alexandra Raikhel) and anti-P53 rabbit polyclonal antibody (ABclonal, cat#: A3185). HRP-conjugated anti-rabbit (Cell Signaling Technology) secondary antibody was used and then detected using Amersham ECL Prime Western Blotting detection reagents. Anti-β-actin-peroxidase antibody (Sigma, cat#: A5316) was used to detect the loading control. All western blotting experiments were repeated at least three times, and representative results are shown in the figures.

IFA for midgut and salivary gland samples followed the previously described protocol[62]. Anti-MAYV E2 antibody (Sigma, cat#: MABF3046) or mouse hyperimmune ascitic fluid specific for DENV2[65] was used to detect MAYV or DENV2 antigen, respectively, and the secondary antibody was AlexaFluor 488 goat anti-mouse IgG (Invitrogen). For TUNEL staining, the TMR red reaction mixture (Roche Applied Science) was incubated with the secondary antibody. DAPI was used to stain the nuclei. Samples were viewed and photographed under a Zeiss LSM700 confocal microscope at the Johns Hopkins University School of Medicine Microscope Facility.

## Caspase activity

Midguts and carcasses were dissected from mosquitoes and homogenized in cold cell lysis buffer from the EnzChek caspase-3 assay kit. After centrifugation at 10,000 g for 10 min at 4 °C, the supernatants were transferred to a new tube. Caspase activity assays were performed using the synthetic fluorescent substrate Z-DEVD-AMC, and the reactions were carried out according to the protocol of the EnzChek caspase-3 assay kit. The fluorescence (excitation/emission, ~342/441 nm) was measured every 5 min for 30 min using a plate reader. At least four technical repeats were performed, with three biological replicates. Data from one biological replicate were used for generating the graph.

## LTR assay

Midguts were dissected from *Ago2* mutant and WT mosquitoes in cold PBS and then stained with 0.1 μM LysoTracker Red DND-99 (Invitrogen) for 5 min. Samples were immediately viewed using under a fluorescence microscope with an excitation/emission of 577/590 nm. Samples were also fixed in 4% paraformaldehyde for 30 min at room temperature. After three washes with PBS, midgut samples were placed individually on a glass slide and mounted with Fluoromount-G with DAPI (Electron Microscopy Sciences Hatfield, PA, USA). Images were obtained using a Zeiss LSM700 confocal microscope.

## Transmission electron microscopy (TEM)

Midguts were dissected and fixed overnight at 4 °C in 2% glutaraldehyde and 2% paraformaldehyde in 0.1 m cacodylate buffer (Sigma Aldrich, St. Louis, MO). Secondary fixation was performed using 1% osmium tetroxide (Ted Pella, Inc. Redding, CA) in 100 mM sodium cacodylate. After dehydration in an ethanol-acetone series, midguts were infiltrated with Epon resin. Midgut sections were cut to a thickness of 85 nm with a diamond knife on the Reichert-Jung Ultracut E ultramicrotome and stained with 2% uranyl acetate in 50% methanol and 0.4% lead citrate. Images were captured at 120 kV on a 16-megapixel CMOS camera on a Talos L120C G2 Transmission Electron Microscope (Thermo-Fisher) at the Johns Hopkins University School of Medicine Microscopy Core.

## RNA-Seq and small RNA library preparation, sequencing, and bioinformatics analysis

Total RNA was extracted from 10 females of MAYV-infected or uninfected WT (Liverpool strain) and *Ago2*$^{-/-}$ mutants at 4 dpi, or DENV2-infected cas9 (parental line) and *AgoN*$^{-/-}$ mutants at 2 and 7 dpi per sample using TRIzol reagent (Invitrogen, Carlsbad, CA), and three independent biological replicates were included for each sample. An Illumina sequencing library was constructed from total RNA using poly(A) enrichment of the mRNA for each RNA sample according to the manufacturer's instructions and sequenced by Novogene Co., LTD (Beijing, China) on the Illumina platform with paired-end 150 bp (PE 150). Data were deposited into NCBI's Sequence Read Archive (SRA) (PRJNA889408). Raw data in FASTQ format were processed to remove reads containing adapters, reads containing ploy-N, and low-quality reads, and clean reads were aligned to the *Ae. aegypti* genome (VectorBase-57). FeatureCounts was used to quantify transcript abundance in each sample using the gene annotation obtained from VectorBase. Differentially expressed (DE) genes between *Ago2*$^{-/-}$ mutants and the WT mosquitoes were identified with DESeq2. Gene ontology (GO) and -fold enrichment of the DE genes were analyzed using the built-in program in Vectorbase or ShinyGO (http://bioinformatics.sdstate.edu/go/). The interaction networks of the DE genes were generated with Revigo (http://revigo.irb.hr/)[66] and modified by Cytoscape[67].

The small RNA libraries were prepared from the total RNA using a Perkin Elmer Small NextFlex v3 according to the standard protocol. The small RNA sequencing was performed on an Illumina NovaSeq6000 with a paired-end 100 bp at the Institute for Genome Sciences, University of Maryland School of Medicine. The paired reads were aligned to the DENV2 or MAYV genome or mosquito virus genome[68] using Geneious Prime or to the *Ae. aegypti* genome (VectorBase-57) using HISAT2 at Usegalaxy (https://usegalaxy.org/).

## Pharmacological treatment, virus injection, and mortality assays

Newly emerged adult females were placed in a cup and were fed with 10% sterile sucrose solution containing 20 mM of apoptosis inducer PAC-1 (Selleck), apoptosis inhibitor Z-VAD-FMK (Selleck), autophagy inducer rapamycin (Millipore), autophagy inhibitor 3-MA (Selleck), or histone deacetylase (HDAC) inhibitor Scriptaid (Millipore). An equivalent amount of DMSO was used as a control. All chemicals were changed daily. On the fifth day of pharmacological treatment, mosquitoes were infected with the various viruses through blood feeding.

The viruses were injected intrathoracically into the mosquitoes by using the Nanoject II injection system (Drummond Scientific) with 69 nl of MAYV ($1.8 \times 10^8$ pfu/ml), DENV2 ($10^7$ pfu/ml) or ZIKV ($10^7$ pfu/ml) per mosquito. An equivalent volume of C6/36 cell culture medium was injected as a control. Each experiment had at least two biological replicates, and each replicate included at least 20 individual females. The number of dead mosquitoes in each cup was recorded daily. Statistical significance was determined by Kaplan-Meier survival analysis with pooled data from different replicates by using GraphPad Prism software, and the *P*-values were determined by the logrank (Mantel-Cox) test.

## Reporting summary

Further information on research design is available in the Nature Portfolio Reporting Summary linked to this article.

# Data availability

The source data for each graph are provided as a Source Data file. The RNA sequencing and small RNA sequencing data generated in this study have been deposited in NCBI's Sequence Read Archive (SRA) under accession number PRJNA889408. Source data are provided with this paper.

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

## Acknowledgements

This work was supported by the National Institutes of Health/National Institute of Allergy and Infectious Disease R01AI141532 (G.D.) and the Bloomberg Philanthropies. The funders had no role in study design, data collection and analysis, decision to publish, or preparation of the manuscript. We thank the Johns Hopkins University School of Medicine Microscope Facility for assistance with confocal imaging and transmission electron microscopy. We also thank Ms. Mihra Tavadia for assistance with bloodfeeding of mosquitoes, the Parasitology Core facilities for providing the naïve human blood used for virus infection, Dr. Omar Akbari at the University of California, San Diego for sharing the germline *Cas9*-expressed *Ae. aegypti* line, Dr. Alexandra Raikhel at the University of California, Riverside for sharing *Ae. aegypti* anti-ATG8 polyclonal antibody, Dr. Kevin Myles at the Texas A & M University for fruitful discussions, and Dr. Deborah McClellan for editorial assistance. The illustrations in Figs. 1, 3, 4 and 7 were created with BioRender.com.

## Author contributions

S.D. Methodology, Data acquisition, Investigation. G.D. Supervision, Funding acquisition. S.D. and G.D. Conceptualization, Data analysis, Writing, editing, and reviewing the manuscript.

## Competing interests

The authors declare no competing interests.
