## [Peer Review File · Nature Communications]

Aedes aegypti Argonaute 2 controls arbovirus infection and host mortalityREVIEWER COMMENTS

Reviewer #1 (Remarks to the Author):

In this manuscript, Dong and Dimopoulos dissect the role of RNAi in antiviral defense in the mosquito *Aedes aegypti*. The authors generate Ago2-deficient mosquitoes to study the effect of the loss of a major antiviral immune pathway on viral replication, and tissue integrity and repair. The mutant mosquitoes showed reduced expression of histones, impaired autophagy and DNA repair, and uncontrolled viral replication *in vivo* leading to hyperinfection and mosquito death. This study is a thorough and structured analysis of the function of the siRNAs pathway as antiviral mechanisms in mosquitoes. Most knowledge on antiviral immunity in Diptera is based on *Drosophila*, however, while many insect-specific viruses often severely impact host survival of flies, arboviruses can establish systemic infections in mosquitoes with little to no pathogenic effects. I am happy to see that the authors performed their experiments in a clean genetic knockout that finally allows for dissecting RNAi-based antiviral immunity in mosquitoes and obtaining clear phenotypes. Though CRSIPR-Cas9 is available in mosquitoes, it is still a non-trivial approach, and therefore I deem this manuscript highly interesting for the field. There are, however, some major difficulties with the description of the knockout system and the analysis of small RNA sequencing data that would need to be resolved before this manuscript can be published.

Major comments:

1) The generated knockout mosquitoes have in-frame mutations and therefore could potentially still have residual activity. The author's observe that siRNA-sized reads are still produced, which is in line with literature, however, it is unclear whether these are functional Ago2-bound siRNAs or not. The authors should show for all their mutants whether Ago2 still has residual activity, or whether Ago2-mediated silencing is fully abolished. This could be achieved for example by one or several of the following: assaying siRNA-mediated target silencing (e.g. *in vivo* RNAi assay similar to Samuel, PNAS 2023 or Merklung, bioRxiv 2022), testing for the presence of mature Ago2-bound siRNAs (e.g. beta-elimination assay), or assessing if Ago2 can still bind siRNAs, for example via IP followed by Northern blotting or sequencing. Residual activity of Ago2 in the different mutants might also explain some of the observed differences between the genotypes.

2) It is unclear for some phenotypes whether they are the result of a different genotype (e.g. uninfected mosquitoes), or a differential response of the genotypes to infection. The fact that the authors observe the biggest differences between mutants and wild types in uninfected conditions, but no or minor additional effects with virus infection for histone expression and DNA repair suggests that the mutants could already show severe phenotypes even in uninfected conditions. The authors should therefore provide at least some basic characterization of the mutants (e.g. fecundity, lifespan, blood-feeding, developmental progression), and uninfected controls for all key experiments (tissue integrity of midguts/salivary glands/malph. tubules, apoptosis, autophagy). Ideally that would also apply to the RNA-seq, but I can see that adding a deep sequencing experiment may be a lot to ask for.

3) The authors do not provide enough details on their sRNA-seq analysis to fully comprehend what they did (see comments below), but I disagree with some of the conclusions the authors drew, and I think that the analysis strategy probably needs to be revised:

a) Comparisons of groups of small RNAs is inherently difficult, because small RNA reads are interdependent, that is, when a class of small RNAs is reduced, this will automatically lead to an increase in sequencing reads of other classes (and vice versa), even when those are not changed. To overcome this problem, the authors should normalize to miRNAs to account for differences in library size, since miRNAs are not expected to be affected in Ago2 knockouts or upon infection. Additionally, the authors should provide experimental evidence to confirm this (e.g. Northern blotting of a few abundant miRNAs).

b) Viral sRNAs should additionally be additionally normalized to viral titers. Since there are so much more viral RNAs and therefore much more substrate for either Dcr2 or the piRNA pathway, the observed increase of siRNA-sized reads is likely solely attributed to an increase in titers. Even more, I would argue that, in contrast to what the authors claim, number of siRNA-sized reads are

actually reduced when compared to degradation products (e.g. reads in the entire range of sRNA sizes, fig. 2b), and I guess that that should become evident when using an appropriate normalization strategy. Additionally, I disagree with the MAYV-reads in the range of 24-31nt to be actual piRNAs, because reads do not exceed that of background levels outside this size range (lack of piRNAs from MAYV is an interesting observation by itself). The authors should provide evidence that the reads are true, mature PIWI-bound piRNAs or otherwise remove the corresponding statements from the manuscript.

c) Fig 2h and the conclusions drawn from it are unclear to me. The authors state that "Ago2 disruption increased the abundance of the mosquito endogenous dsRNA, as reflected in the upregulation of noncoding RNA (ncRNA)", however, I do not understand how the authors assessed increased dsRNA formation from their data. Non-coding RNAs do not equal dsRNAs, and the authors do not provide any information on how they extracted this particular information.

Minor comments

- 1) The authors use the parental Cas9 line (that was used to generate the mutants and for the first crosses) as control for their experiments in the beginning of the manuscript, but switch to the Liverpool strain (used for back-crossing) for later experiments. I guess the choice for any of the two lines as control is valid, but the authors should show that both lines behave similarly in their assays and can indeed be used as controls interchangeably.
- 2) Additionally, though the presented IFA images are convincing, quantifications from several images should be included to provide a quantitative estimation of effect sizes.
- 3) The authors should report exact numbers on how often an experiment was repeated, and how many mosquitoes/tissues/etc were used for each experiment (in the figures themselves or in the legends).
- 4) The authors mentioned ArgoN^{-/-}2 mutants, but do not show any data for this mutant line. If they did not perform any experiments with this mutant they could delete it from the manuscript.
- 5) Some of the figures, especially IFA and EM images, are very small. I would advise the authors to enlarge these images to make it easier to see details. Additionally, sometimes the authors present images with very different scales for each image which makes them hard to compare.
- 6) Fig2e, S2e: I suggest to split the plot into reads on +/- strand and siRNA-sized and piRNA-sized reads to be able to see exact read coverage along the viral genome.
- 7) The method section for deep sequencing experiments and analyses are insufficiently described to reproduce the findings. E.g. did the authors perform polyA-enrichment or rRNA-depletion? Which kits/reagents were used? Which mosquitoes (genotypes, conditions) were used? Does wild-type refers to Liverpool strain or Cas9 line?
- 8) The authors should include the DE-seq output in tabular form as supplementary material.
- 9) PCA: this method identifies principal components that can explain variation in data. However, the authors labeled the axes titer vs sRNA counts. The authors should make sure that what they present is actually a PCA and not a simple scatterplot.
- 10) Fig. 3b-e: are these blood-fed or injected mosquitoes? Please provide this information.
- 11) Fig 4j: it is not clear which genes these names refer to (I tried to find them in VectorBase but was not successful). The authors should provide gene identifiers in the manuscript for the analyzed genes to reduce ambiguity.
- 12) "Arbovirus-induced mortality was also observed in Dicer2^{-/-} mutant mosquitoes that impair the siRNA pathway, but mortality in Dicer2^{-/-} mutants is lower than that in Ago2^{-/-} mutants, indicating Ago2 may play other roles in protecting mosquito death caused by arbovirus infection". However, in Samuel, PNAS 2023 (<https://doi.org/10.1073/pnas.2213701120>) the Dcr2^{-/-} mutants show severe lethality upon infection, which is in contrast to the study the authors cite (Merkling 2022). I would argue that lethality in Dcr2 KO mosquitoes is not conclusive so far, and comparison with the Ago2 mutants not directly possible without additional experiments including both genotypes. Therefore, I would suggest to adjust the statement that Ago2 knockouts show different phenotypes than what can be observed from loss of siRNA pathway in general.
- 13) Readability of the manuscript could be enhanced by adding sub-headings for each section.
- 14) The figure description should be revised to include all important experimental details necessary to understand the experiment and analysis.
- 15) The manuscript contains some minor typos and grammar mistakes. Please should correct those.
- 16) RNAseq data should be made publicly available prior to publication.

Reviewer #2 (Remarks to the Author):

In the manuscript, Ago2^{-/-} mutant mosquitoes are generated and investigated for their altered response against arbovirus infections (mainly MAYV and DENV2). As expected, much higher levels of virus infection were observed but the new study differs from previous ones by trying to elucidate if alternative immune pathways besides RNAi become activated in Ago2^{-/-} mosquitoes. More specifically, increased apoptosis was observed but also a decrease in DNA repair and autophagy which was correlated with decreased histone expression. An interesting observation is also that, despite higher viral replication in tissues, Ago2^{-/-} mutant mosquitoes were less capable to discharge new viral particles during feeding (also related to increased tissue damage). The novelty of the study lays at the clarification of possible other roles that Ago2 could play in antiviral defense beyond its well-known function in the siRNA pathway.

General comments:

It can be assumed that Ago2 in mosquitoes does not only control virus infection but also other parasitic elements such as transposons. A major overlook in the presented work is whether mobilization of transposons is altered in Ago2^{-/-} mutant mosquitoes. An increase in transposon mobilization would result in increased DNA damage which would be exacerbated by the decrease in expression of DNA repair genes. However, increased transposon expression would also result in increased inflammation and immune gene expression, even in mosquitoes that are not infected. This issue may be very important and it is therefore recommended to carry out RNA-seq on Ago2^{-/-} mutant mosquitoes that are not infected with arbovirus.

The authors claim the importance of the regulation of DNA repair, autophagy and histone genes by Ago2 in their study. However, all these effects may be indirect because the absence of Ago2 will result in high levels of virus replication that causes large damage to tissues followed by inflammation and stress responses. The authors should take into account seriously that the effects on gene expression are the result of the stress/inflammation response caused by damage. Similarly, knockdown of histones is expected to affect gene expression on a global genome scale with possible activation of parasitic elements/transposons that can dysregulate defense pathways such as DNA repair, apoptosis and autophagy.

Other comments:

The results that are presented in Figure 1 are not described well in the text. To which part of Figure 1 do lines 122-126 refer to (Extended Figure 1F is irrelevant)? In Fig. 1g, the number of samples of Ago2^{-/-} mosquitoes in the graph that measures pfu/mL seems much lower than indicated in the graph next to it (infected %). In general, the samples of Ago2^{-/-} mosquitoes are much smaller than WT mosquitoes. In Fig. 1e and 1f, it should be indicated that the midgut epithelium is infected. In Extended Fig. 1d, other tissues besides midgut epithelium (tracheae, muscle?) need to be indicated.

Line 146: "evenly distributed" does not seem to be correct as there is a clear peak of 21 nt (Fig 2b). What is clear is that the "background" reads of the small RNAs is considerably higher in Ago2^{-/-} than in WT – this is a clear difference (Fig 2b). "Evenly distributed" applies to Fig. 2e, not 2b. When the authors talk about vpiRNAs, they should describe how the vpiRNAs were determined. On the basis of length, it cannot be decided. In that case, the term vpiRNA-like sRNAs should be used but of course it is important to determine to what extent bona fide vpiRNAs are produced (e.g. ping-pong amplification).

In Extended Fig. 2c, is the scale in the Y-axis correct? Why is there no peak at 21 nt?

Line 203: "strong fluorescence with a "ball"-like shape": the ball-like shape have midgut cells, not the midgut?

Line 220: "clusters of virions were observed in the midguts" should be "in the midgut cells"?

Staining of midgut epithelial cells: only staining for viral antigen and DAPI is shown. It would be useful to stain for typical markers in the epithelial cells to reveal cell shape and cell identity.

Line 313: "These data indicate that either silencing of histone mRNAs or inhibiting histone modification adversely affects arbovirus replication in the mosquitoes and renders them more susceptible to arboviral infection.": re-formulate; silencing of histones results in a decrease in virus titer.

Line 383: "mortality in Dicer2^{-/-} mutants is lower than that in Ago2^{-/-} mutants": provide quantitative comparison.

Line 406: "other conserved immune signaling cascades, including the Toll and apoptosis pathways, are induced upon virus infection and may play a role in defending against viral infection": the virus may not be the primary trigger of the immune cascades but instead the damage inflicted by

uncontrolled virus replication.

Line 417: "Ago2 disruption does not affect the endonuclease activity of Dcr2": this is also known in *Drosophila* (provide reference).

Reviewer #3 (Remarks to the Author):

In the present manuscript by Dong and Dimopoulos, the authors describe a new Ago-2 mutated mosquito model and its impacts on arbovirus infection and mosquito survival after infection. The manuscript includes a lot of valuable information and a new gene knockout model to study various antiviral responses and mechanisms, but there are some major concerns that need to be addressed in this reviewer's opinion.

1) The authors describe the Ago-2 mutant that has a 72 bp deletion as an Ago2 knockout mosquito. I am not sure if this is an entirely fair representation. It is a deletion mutant, lacking part of the protein, but if the rest of the protein is still made, I am hesitant to accept the term 'knockout'.

2) The authors do not show a dysfunction in the RNAi pathway using for example dsRNA silencing of an endogenous gene. Now with the relevant deletion in the Ago domain, a lack of slicing activity is implied, but evidence of it would have been good to show (e.g. parallel injections of wt and mutant mosquitoes with dsRNA, validation of silencing in wt, lack of silencing in mutant)

3) It is hard to get overly excited about the histone/DNA repair/autophagy axis considering the immense impact that high levels of replication appear to have on the cells. Is histone downregulation truly the 'first' issue or is it merely a by-product of impaired DDR/other cellular damage? The causes and consequences of Ago2 loss of function and increased virus replication are hard to discern for sure from the present study.

4) There were a lot of discrepancies with reference to figures and content of figures as described below that made it hard to follow at times. Please proof-read extensively.

Specific comments:

Line 126: In the files provided to this reviewer, no 'Extended Figure 1f' was present. Extended Figure 1 ends with Extended Figure 1D. The paper also does not reference any Extended Figure 1E. Please double check what happened there.

Line 129: Ok you then reference Figure 1g for the count of female mosquitoes which fed, so I assume the previous reference to Extended Figure 1D, was intended for Figure 1g. Seems like the reference to 1h also needs to be to 1i and the one currently saying 1i needs to be 1jj. Please fix these.

Figure Legend 1H - 'percentage of mosquitoes in each cup' - without further context, this is not a sufficient legend and it is unclear what is shown unless you find the relevant section in the text. It may not be required, but it would be ideal if the reader can understand the figure+legend on its own.

Line 159: The authors state that 'Therefore, our results show that Ago2 disruption does not affect the production of arboviral sRNAs but does impair the ability of the siRNA pathway to degrade viral RNAs.' Where do the authors show that degradation of viral RNA is impaired? The authors show an increase in viral RNA and a similar abundance of sRNAs (relatively per genome copy), but how can the authors conclude that the sRNA response is inhibited at this stage? (Aside from drawing from their hypothesis). While I don't disagree that this is highly likely the case, the data does not provide any evidence at this point. Only a functional assay measuring Ago-2 slicing activity would provide this direct evidence.

Figure 4g ◊ the MAYV infected Ago2 knockout midguts overlay image appears to be missing the TUNEL channel in the overlay. The individual channels are shown, but the overlay shows no 'red', while this is clearly visible in the wildtype cell overlay.

Line 278 – typo, should be 'involved' with a d at the end

Line 304: The authors state: 'In the WT mosquitoes, MAYV or DENV2 infection significantly reduced the expression of histone genes when compared to non-infected mosquitoes'. For MAYV in Fig 5i, there is no evidence that a direct mock vs MAYV condition showed significantly reduced histone gene expression. All shown significance bars/asterisks span from one mosquito group to the other, not simply between wt mock and wt MAYV. Having said that, the Extended Figure 5b seems to show an impact of MAYV infection on histone 4. However, it is unclear how that figure differs from Figure 5i (experimentally – both say 4 days post MAYV infection. Are they replicates??).

Going back to the original statement in the paper, DENV infection (Extended Figure 5c) if anything increased expression of histone genes in wildtype mosquitoes. Please fix this entire section, the references to figures and make the shown results clearer.

Lines 310-315: When the authors introduce the HDAC inhibitor, it might be better to provide a bit more context for a broad audience. The authors simply say 'inhibiting histone modification adversely affects arbovirus replication'. HDAC inhibition would directly result in an increase in euchromatin formation and prevent compact/strong histone-DNA interaction. If a HAT inhibitor was used, the opposite results may occur. So simply labeling it 'histone modification' is a bit simplistic here.

Figure 6a shows the 3 genes, p16, Atg2 and Atg6, but the text refers to p62 and two Atgs. Please correct/clarify as needed (e.g. was p16 just a typo and it is p62?).

Line 328: the authors refer to the confirmation of 'the upregulation of p62 in Ago2^{-/-} mutants with or without MAYV infection at 4 dpi'. However, the figure shows that p62 is downregulated in uninfected Ago2 knockout mosquitoes and then upregulated back to the wt levels following infection. Furthermore the authors refer to changes in these genes (p62, Atgs) in DENV infected mosquitoes in Extended Figure 6a, but extended Figure 6a is a Western Blot of ATG-8 specifically.

Line 359, the authors refer to Figure 4J, but it is Figure 6J.

Line 360 – I believe (?) the correct figure to reference here is Extended Figure 6f . Please check all these references and figure panels.

Line 360-362: This is again slightly inaccurate – the authors state that 'As compared to the sugarfed mosquitoes, the bloodfed mosquitoes exhibited a significantly higher expression of autophagy genes in the GFP control mosquitoes, but not in the histone-silenced mosquitoes.' However, in the referenced Figure 6, expression of two of the autophagy genes (Atg-7 and Atg-12) does increase when comparing between the histone-silenced sugarfed mosquitoes and the histone-silenced bloodfed mosquitoes – just not to the same degree as in the GFP control. Please make sure you are accurate in your descriptions.

The model in Figure 7 is labeled 'miRNA' as the small RNA in Ago-2 ... this should obviously be siRNA.

Point-by-point response to reviewer comments

We thank the reviewers for insightful comments which we have addressed by performing additional experiments and making appropriate changes in the text, thereby significantly improving the quality of our manuscript.

REVIEWER #1

GENERAL COMMENT: In this manuscript, Dong and Dimopoulos dissect the role of RNAi in antiviral defense in the mosquito *Aedes aegypti*. The authors generate Ago2-deficient mosquitoes to study the effect of the loss of a major antiviral immune pathway on viral replication, and tissue integrity and repair. The mutant mosquitoes showed reduced expression of histones, impaired autophagy and DNA repair, and uncontrolled viral replication in vivo leading to hyperinfection and mosquito death.

This study is a thorough and structured analysis of the function of the siRNAs pathway as antiviral mechanisms in mosquitoes. Most knowledge on antiviral immunity in Diptera is based on *Drosophila*, however, while many insect-specific viruses often severely impact host survival of flies, arboviruses can establish systemic infections in mosquitoes with little to no pathogenic effects. I am happy to see that the authors performed their experiments in a clean genetic knockout that finally allows for dissecting RNAi-based antiviral immunity in mosquitoes and obtaining clear phenotypes. Though CRISPR-Cas9 is available in mosquitoes, it is still a non-trivial approach, and therefore I deem this manuscript highly interesting for the field.

There are, however, some major difficulties with the description of the knockout system and the analysis of small RNA sequencing data that would need to be resolved before this manuscript can be published.

RESPONSE: We thank the reviewer for positive feedback; we have addressed all comments accordingly, by making revisions in the text and performing additional experiments.

Major comments:

COMMENT 1: The generated knockout mosquitoes have in-frame mutations and therefore could potentially still have residual activity. The authors observe that siRNA-sized reads are still produced, which is in line with literature, however, it is unclear whether these are functional Ago2-bound siRNAs or not. The authors should show for all their mutants whether Ago2 still has residual activity, or whether Ago2-mediated silencing is fully abolished. This could be achieved for example by one or several of the following: assaying siRNA-mediated target silencing (e.g. in vivo RNAi assay similar to Samuel, PNAS 2023 or Merklings, bioRxiv 2022), testing for the presence of mature Ago2-bound siRNAs (e.g. beta-elimination assay), or assessing if Ago2 can still bind siRNAs, for example via IP followed by Northern blotting or sequencing. Residual activity of Ago2 in the different mutants might also explain some of the observed differences between the genotypes.

RESPONSE: According to the literature and our previous study (Olmo et al. 2018 and Dong et al. 2023), *Ago2* is downstream of *Dicer2* that cleaves dsRNA into siRNAs and knockout of *Ago2* should not impact function of these siRNAs. To test whether *Ago2*-mediated silencing is fully abolished, we performed three independent experiments of silencing various endogenous genes in WT and *Ago2*^{-/-} mutants. 1) we silenced *kynurenine hydroxylase (kh)* gene (AAEL008879) in newly laid eggs of WT and *Ago2* knockout mutants through microinjection of *kh* dsRNA. The control eggs were injected with dsGFP. The *kh* gene targets kynurenine 3-monooxygenase (KMO) function critical for eye pigmentation and knockout of *kh* leads to severe eye pigmentation defects (Basu et al. 2015). As compared to the dsGFP-injected control, a serious defect in eye pigmentation was observed in 3rd instar larvae hatched from *dkh*-injected WT eggs, however, we did not observe defect in eye pigmentation in larvae hatched from *dkh*-injected *Ago2*^{-/-} eggs (**Revised Fig. 1a**). We observed a mild defect in eye pigmentation in larvae hatched from *dkh*-injected *AgoN*^{-/-} eggs. These results indicated *Ago2*^{-/-} mutant is functionally defective of the siRNA, and is therefore a *Ago2*^{-/-} null mutant. The *AgoN*^{-/-} mutant still has some residual siRNA activity and can therefore be considered a partially *Ago2*^{-/-} knockout mutant.

2) we silenced the *eggshell organizing factor 1 (EOF1)* gene (AAEL012336) in WT and *Ago2*^{-/-} females through injection of ds*EOF1*. The control females were injected with dsGFP. *EOF1* plays an essential role in the formation and melanization of the eggshell, and silencing of *EOF1* leads to a defect in eggshell melanization and reduction in fecundity (Isoe et al. 2019). ds*EOF1*-injected WT females produced significantly lesser eggs and those few eggs produced had defect in their melanization when compared to dsGFP-injected control. The ds*EOF1*-injected *Ago2*^{-/-} females did not show any difference in fecundity and eggshell melanization as compared to the dsGFP-injected controls (**Revised Fig. 1b**).

3) we silenced three endogenous genes (*AeIMPDH*, *AeCRVP* and *AeTry196*) that have been reported to show significant silencing and/or phenotypes in *Ae. aegypti* in our previous studies (Angleró-Rodríguez et al. 2017 and Tikhe et al. 2022). Compared to WT where gene silencing is significant, *Ago2*^{-/-} mutants didn't show gene silencing (**Revised Fig. 1c**).

Taken together, our data strongly support that *Ago2* function in silencing is fully abolished in *Ago2*^{-/-} mutants. See figure and references below.

Revised Fig.1 Characterization of *Ago2* knockout mutants. **a**, phenotypes of 3rd instar larvae hatched from *dsGFP*-injected eggs of WT, and *Ago2*^{-/-} and *ArgoN*^{-/-} mutants. Larvae hatched from *dsGFP*-injected eggs were used as a control. Yellow arrows indicate defect in eye pigmentation. **b**, eggs laid from *dsEOF1*-injected females of WT and *Ago2*^{-/-} mutants. P-values were determined by using an unpaired *t*-test. *****P*<0.0001. **c**, silencing efficiency of endogenous genes (*AeIMPDH*, *AeCRVP* and *AeTry196*) in WT and *Ago2*^{-/-} mutants at 3 days post their dsRNA injection. Statistical significance was determined by a one-way ANOVA. **P*<0.05, ***P*<0.01, ****P*<0.001.

References

Angleró-Rodríguez YI, Talyuli OA, Blumberg BJ, et al. An *Aedes aegypti*-associated fungus increases susceptibility to dengue virus by modulating gut trypsin activity. *Elife*. 2017;6:e28844.

Basu S, Aryan A, Overcash JM, et al. Silencing of end-joining repair for efficient site-specific gene insertion after TALEN/CRISPR mutagenesis in *Aedes aegypti*. *Proc Natl Acad Sci U S A*. 2015;112(13):4038-4043.

Dong Y, Dong S, Dizaji NB, et al. The *Aedes aegypti* siRNA pathway mediates broad-spectrum defense against human pathogenic viruses and modulates antibacterial and antifungal defenses *PLoS Biol*.

2022;20(7):e3001722.

Isoe J, Koch LE, Isoe YE, et al. Identification and characterization of a mosquito-specific eggshell organizing factor in *Aedes aegypti* mosquitoes. *PLoS Biol*. 2019;17(1):e3000068.

Olmo RP, Ferreira AGA, Izidoro-Toledo TC, et al. Control of dengue virus in the midgut of *Aedes aegypti* by ectopic expression of the dsRNA-binding protein Loqs2. *Nat Microbiol*. 2018;3(12):1385-1393.

Tikhe CV, Cardoso-Jaime V, Dong S, Rutkowski N, Dimopoulos G. Trypsin-like Inhibitor Domain (TIL)-Harboring Protein Is Essential for *Aedes aegypti* Reproduction. *Int J Mol Sci*. 2022;23(14):7736.

COMMENT 2: It is unclear for some phenotypes whether they are the result of a different genotype (e.g. uninfected mosquitoes), or a differential response of the genotypes to infection. The fact that the authors observe the biggest differences between mutants and wild types in uninfected conditions, but no or minor additional effects with virus infection for histone expression and DNA repair suggests that the mutants could already show severe phenotypes even in uninfected conditions. The authors should therefore provide at least some basic characterization of the mutants (e.g. fecundity, lifespan, blood-feeding, developmental progression), and uninfected controls for all key experiments (tissue integrity of midguts/salivary glands/maliph. tubules, apoptosis, autophagy). Ideally that would also apply to the RNA-seq, but I can see that adding a deep sequencing experiment may be a lot to ask for.

RESPONSE: As suggested by the reviewer, we compared some basic fitness parameters (e.g., adult longevity, blood-feeding capacity, larva development, adult body weight and fecundity/fertility) between uninfected WT and *Ago2*^{-/-} mutants. Our results show that there are no differences in whole-body weight (**Revised Fig. 2a**), blood-feeding percentage (**Revised Fig. 2b**), amount of blood taken (**Revised Fig. 2c**) and fertility between WT and *Ago2*^{-/-} mutants (**Revised Fig. 2f**), however, *Ago2*^{-/-} mutants have a significantly delayed larva development (**Revised Fig. 2d**), a reduced fecundity (**Revised Fig. 2e**) and a shorter life span of females and males (**Revised Fig. 2g**). See figure below.

We also performed IFA on midguts, ovaries and salivary glands of uninfected WT and *Ago2*^{-/-} mutants to check cell lysis/tissue integrity. Compared to the WT, we did not observe lysis of these tissues in uninfected *Ago2*^{-/-} mutants (**Revised Fig. 3**). See figure below.

We also performed RNA-seq on uninfected WT and *Ago2*^{-/-} mutants at 4 days post blood feeding. The results showed that there were 555 upregulated and 280 downregulated differentially expressed (DE) genes in uninfected *Ago2*^{-/-} mutants as compared to uninfected WT (**supplementary Table 2** in manuscript), suggesting *Ago2* disruption has a significant impact on gene expression. Further analysis revealed that mRNA abundance of most of the histone genes were reduced in *Ago2*^{-/-} mutants (**Revised Fig.4a** and **supplementary Table 2** in manuscript), and overall histone mRNA abundance was significantly decreased in *Ago2*^{-/-} mutants by 24.59% (**Revised Fig.4b**). The significantly down-regulated histone genes include four *Histone 2A*, two *Histone 2B*, two *Histone 3*, two *Histone 4* and one H1/H5 linker (**Revised Fig.4c**). mRNA abundance of DNA repair genes and autophagy genes were also reduced in uninfected *Ago2*^{-/-} mutants as compared to uninfected WT (**Revised Fig.4d,e** and **supplementary Table 2** in manuscript). However, a number of the immunity genes were increased in uninfected *Ago2*^{-/-} mutants as compared to uninfected WT (**Revised Fig.4f** and **supplementary Table 2** in manuscript). We did not compare virus infected vs uninfected *Ago2*^{-/-} mutants or WT mosquitoes because we did RNA-seq on infected and uninfected at different times and different batches of mosquitoes, which may be affected by library preparation, sequencing depth, etc. See figure below.

COMMENT 3: The authors do not provide enough details on their sRNA-seq analysis to fully comprehend what they did (see comments below), but I disagree with some of the conclusions the authors drew, and I think that the analysis strategy probably needs to be revised:

COMMENT 3a. Comparisons of groups of small RNAs is inherently difficult, because small RNA reads are interdependent, that is, when a class of small RNAs is reduced, this will automatically lead to an increase in sequencing reads of other classes (and vice versa), even when those are not changed. To overcome this problem, the authors should normalize to miRNAs to account for differences in library size, since miRNAs are not expected to be affected in *Ago2* knockouts or upon infection. Additionally, the authors should provide experimental evidence to confirm this (e.g. Northern blotting of a few abundant miRNAs).

RESPONSE: We agree that small RNA reads are interdependent (as shown in **Fig.2h**, where *Ago2*^{-/-} mutants has high number of MAYV vsRNA reads and low number of mosquitoes sRNA reads as compared to WT mosquitoes). When we analyzed viral (MAYV and DENV2) sRNA reads, we normalized to the total sRNA counts for each library to reduce difference of library size (total sRNA sequencing reads per sample) and presented as reads per million total reads (RPM). Following the reviewer's suggestion, we analyzed the abundance of total mosquito miRNAs for each library. Because each sRNA sequencing yields different total sRNA reads, we normalized miRNAs to the total reads of each library. The results showed that there were no differences of normalized counts to total miRNAs between WT and *Ago2*^{-/-} mutants (**Revised Fig.5a**), suggesting *Ago2* disruption may not affect the overall abundance of miRNAs in mosquitoes. We also normalized MAYV sRNA counts to the total miRNA counts for each library, and results showed the ratio of MAYV sRNA counts to total miRNA counts are 36-fold higher in *Ago2*^{-/-} mutants than that in WT (**Revised Fig.5b**), which is similar to result of MAYV sRNA counts normalized to the total sRNA reads (**Fig.2a** in manuscript). Therefore, using viral sRNA reads normalized to the library size can reflect the difference of viral sRNAs between WT and *Ago2*^{-/-} mutants. See figure below.

COMMENT 3b. Viral sRNAs should additionally be additionally normalized to viral titers. Since there are so much more viral RNAs and therefore much more substrate for either Dcr2 or the piRNA pathway, the observed increase of siRNA-sized reads is likely solely attributed to an increase in titers. Even more, I would argue that, in contrast to what the authors claim, number of siRNA-sized reads are actually reduced when compared to degradation products (e.g. reads in the entire range of sRNA sizes, fig. 2b), and I guess that that should become evident when using an appropriate normalization strategy. Additionally, I disagree with the MAYV-reads in the range of 24-31nt to be actual piRNAs, because reads do not exceed that of background levels outside this size range (lack of piRNAs from MAYV is an interesting observation by itself). The authors should provide evidence that the reads are true, mature PIWI-bound piRNAs or otherwise remove the corresponding statements from the manuscript.

RESPONSE: We agree that increase of viral siRNA is attributed to an increase in viral titer in *Ago2*^{-/-} mutants, as that we present in **Fig.2b** and **Extended Data Fig.4k**. We calculated the ratio of 21-nt viral siRNA to the all size (18nt to 40nt) of viral sRNAs, and results showed that the ratio of 21-nt viral siRNA is significantly decreased (**Revised Fig.6**) in *Ago2*^{-/-} mutants, and we revised the manuscript accordingly. As suggested, we removed results of piRNAs as the manuscript mainly focuses on the siRNA pathway. See figure to the right.

COMMENT 3c. Fig 2h and the conclusions drawn from it are unclear to me. The authors state that “Ago2 disruption increased the abundance of the mosquito endogenous dsRNA, as reflected in the upregulation of noncoding RNA (ncRNA)”, however, I do not understand how the authors assessed increased dsRNA formation from their data. Non-coding RNAs do not equal dsRNAs, and the authors do not provide any information on how they extracted this particular information.

RESPONSE: We agree, non-coding RNAs do not equal dsRNAs and long non-coding RNAs (lncRNAs) are single-stranded RNAs. We have removed data regarding ncRNAs since we mainly focus on the siRNA pathway.

Minor comments (reviewer 1)

MINOR COMMENT 1. The authors use the parental Cas9 line (that was used to generate the mutants and for the first crosses) as control for their experiments in the beginning of the manuscript, but switch to the Liverpool strain (used for back-crossing) for later experiments. I guess the choice for any of the two lines as control is valid, but the authors should show that both lines behave similarly in their assays and can indeed be used as controls interchangeably.

RESPONSE: The parental Cas9 line (*exu-Cas9*) was generated from the Liverpool strain (LVP) (Li et al. 2017). We performed viral infection assays on these two mosquito strains, and we did not observe any difference regarding MAYV and DENV2 infection (**Revised Fig.7**).

See figure to the right and reference below.

Reference:

Li, M. et al. Germline Cas9 expression yields highly efficient genome engineering in a major worldwide disease vector, *Aedes aegypti*. P Natl Acad Sci USA 114, E10540-E10549, doi:10.1073/pnas.1711538114 (2017).

MINOR COMMENT 2. Additionally, though the presented IFA images are convincing, quantifications from several images should be included to provide a quantitative estimation of effect sizes.

RESPONSE: We did not quantitatively estimate the virus-lysed cells in midguts of *Ago2*^{-/-} mutants since there are no lysed cells observed in WT midguts after virus infection. But we did count autophagy cells from IFA images, as presented in **Fig.6d**.

MINOR COMMENT 3. The authors should report exact numbers on how often an experiment was repeated, and how many mosquitoes/tissues/etc were used for each experiment (in the figures themselves or in the legends).

RESPONSE: We included this information in Materials and Methods, but we also added this information in the figure legends as suggested.

MINOR COMMENT 4. The authors mentioned *ArgoN*^{-/-2} mutants, but do not show any data for this mutant line. If they did not perform any experiments with this mutant they could delete it from the manuscript.

RESPONSE: We deleted the *ArgoN*^{-/-2} mutants in the revised manuscript.

MINOR COMMENT 5. Some of the figures, especially IFA and EM images, are very small. I would advise the authors to enlarge these images to make it easier to see details. Additionally, sometimes the authors present images with very different scales for each image which makes them hard to compare.

RESPONSE: We have enlarged some images and changed scales for some images. Because images were taken at different magnification, it is difficult to use one scale for all the images. We usually presented an overview image of midgut or salivary gland, and also present the detail of interested by highlighting with white box line.

MINOR COMMENT 6. Fig2e, S2e: I suggest to split the plot into reads on +/- strand and siRNA-sized and piRNA-sized reads to be able to see exact read coverage along the viral genome.

RESPONSE: To increase the accuracy of viral small RNA, we used pair-end sequencing for all the small RNA libraries. Therefore, it would not differentiate +/- strand for the pair-end sequencing data. However, we aligned one of the pair-end sequencing data (_R1) to the DENV2/MAYV genome, and the results showed that 21nt viral siRNA coverage along the MAYV or DENV2 genome is very similar between WT and *Ago2*^{-/-} mutants, but *Ago2*^{-/-} mutants have a greater siRNA abundance than WT (**Revised Fig.8**). We also added size distribution of viral sRNA mapped to positive and negative strands of MAYV and DENV2 (**Extended Data Fig.4b,d,e** in manuscript), respectively. See figure below.

Revised Fig.8 Distribution and the relative abundances of 21-nt viral siRNAs in WT and *Ago2* knockout mutants. **a**, distribution of 21-nt siRNAs in positive and negative strand of the MAYV genome between WT and *Ago2*^{-/-} mutants at 4 days post MAYV infection. **b**, distribution of 21-nt siRNAs in positive and negative strand of the DENV2 genome between *Cas9* and *ArgoN*^{-/-} mutants at 7 days post DENV2 infection.

MINOR COMMENT 7. The method section for deep sequencing experiments and analyses are insufficiently described to reproduce the findings. E.g. did the authors perform polyA-enrichment or rRNA-depletion? Which kits/reagents were used? Which mosquitoes (genotypes, conditions) were used? Does wild-type refers to Liverpool strain or *Cas9* line?

RESPONSE: All the RNA-Seq libraries were prepared from total RNA using poly(A) enrichment of the mRNA and sequenced by Novogene (mRNA-Seq). All the small RNA sequencing libraries were prepared using a Perkin Elmer Small NextFlex v3 and sequenced at the Institute for Genome Sciences, University of Maryland School of Medicine. The mosquitoes (genotypes and infection) were added in the method section of the revised manuscript. The wild-type refers to Liverpool (LVP) strain, and *Cas9* line was used for a parental line for *ArgoN*^{-/-} mutants.

MINOR COMMENT 8. The authors should include the DE-seq output in tabular form as supplementary material.

RESPONSE: We added upregulated and downregulated DE genes in **Table S1**.

MINOR COMMENT 9. PCA: this method identifies principal components that can explain variation in data. However, the authors labeled the axes titer vs sRNA counts. The authors should make sure that what they present is actually a PCA and not a simple scatterplot.

RESPONSE: We used a PCA analysis from the GraphPad software.

MINOR COMMENT 10. Fig. 3b-e: are these blood-fed or injected mosquitoes? Please provide this information.

RESPONSE: These were blood-fed mosquitoes. We added this information in the figure legend.

MINOR COMMENT 11. Fig 4j: it is not clear which genes these names refer to (I tried to find them in VectorBase but was not successful). The authors should provide gene identifiers in the manuscript for the analyzed genes to reduce ambiguity.

RESPONSE: We updated the gene names and their corresponding vector IDs in **Table S1, S2** and **S8**, as well as in the manuscript.

MINOR COMMENT 12. "Arbovirus-induced mortality was also observed in *Dicer2*^{-/-} mutant mosquitoes that impair the siRNA pathway, but mortality in *Dicer2*^{-/-} mutants is lower than that in *Ago2*^{-/-} mutants, indicating *Ago2* may play other roles in protecting mosquito death caused by arbovirus infection". However, in Samuel, PNAS 2023

(<https://doi.org/10.1073/pnas.2213701120>) the *Dcr2*^{-/-} mutants show severe lethality upon infection, which is in contrast to the study the authors cite (Merkling 2022). I would argue that lethality in *Dcr2* KO mosquitoes is not conclusive so far, and comparison with the *Ago2* mutants not directly possible without additional experiments including both genotypes. Therefore, I would suggest to adjust the statement that *Ago2* knockouts show different phenotypes than what can be observed from loss of siRNA pathway in general.

RESPONSE: We thank the reviewer for pointing this out. When we submitted our manuscript, Samuel et al. 2023 PNAS was not published. Therefore, we did not include discussion about their *Dcr2*^{-/-} mutants, but we now rephrased this statement as suggested (lines 447-456).

MINOR COMMENT 13. Readability of the manuscript could be enhanced by adding sub-headings for each section.

RESPONSE: We added sub-headings for some sections when relevant.

MINOR COMMENT 14. The figure description should be revised to include all important experimental details necessary to understand the experiment and analysis.

RESPONSE: We updated this information accordingly.

MINOR COMMENT 15. The manuscript contains some minor typos and grammar mistakes. Please should correct those.

Response: We have corrected some typos.

MINOR COMMENT 16. RNAseq data should be made publicly available prior to publication.

RESPONSE: Yes, we already deposited our deep sequencing data (SRA: PRJNA889408) and will make them publicly available if the manuscript is accepted for publication.

REVIEWER #2 (Remarks to the Author):

In the manuscript, Ago2^{-/-} mutant mosquitoes are generated and investigated for their altered response against arbovirus infections (mainly MAYV and DENV2). As expected, much higher levels of virus infection were observed but the new study differs from previous ones by trying to elucidate if alternative immune pathways besides RNAi become activated in Ago2^{-/-} mosquitoes. More specifically, increased apoptosis was observed but also a decrease in DNA repair and autophagy which was correlated with decreased histone expression. An interesting observation is also that, despite higher viral replication in tissues, Ago2^{-/-} mutant mosquitoes were less capable to discharge new viral particles during feeding (also related to increased tissue damage). The novelty of the study lays at the clarification of possible other roles that Ago2 could play in antiviral defense beyond its well-known function in the siRNA pathway.

General comments:

COMMENT 1. It can be assumed that Ago2 in mosquitoes does not only control virus infection but also other parasitic elements such as transposons. A major overlook in the presented work is whether mobilization of transposons is altered in Ago2^{-/-} mutant mosquitoes. An increase in transposon mobilization would result in increased DNA damage which would be exacerbated by the decrease in expression of DNA repair genes. However, increased transposon expression would also result in increased inflammation and immune gene expression, even in mosquitoes that are not infected. This issue may be very important and it is therefore recommended to carry out RNA-seq on Ago2^{-/-} mutant mosquitoes that are not infected with arbovirus.

RESPONSE: Following the reviewer's suggestion, we performed RNA-seq on uninfected WT and Ago2^{-/-} mutants at 4 days post blood feeding, and the results were presented in **Revised Fig.4** (see figure above in our response to reviewer 1) and **supplementary Table 2** in the manuscript. We observed histone loss and decrease in expression of DNA repair genes and autophagy genes and increase in immune gene expression (most of them with predicated antibacterial function) in uninfected Ago2^{-/-} mutants as compared to uninfected WT. These data support the hypothesis that Ago2 has other functions beyond a key component of the siRNA pathway to control arbovirus infection in mosquitoes, and we have discussed this in the manuscript (lines 442-447, 500-510).

COMMENT 2. The authors claim the importance of the regulation of DNA repair, autophagy and histone genes by Ago2 in their study. However, all these effects may be indirect because the absence of Ago2 will result in high levels of virus replication that causes large damage to tissues followed by inflammation and stress responses. The authors should take into account seriously that the effects on gene expression are the result of the stress/inflammation response caused by damage.

Similarly, knockdown of histones is expected to affect gene expression on a global genome scale with possible activation of parasitic elements/transposons that can dysregulate defense pathways such as DNA repair, apoptosis and autophagy.

RESPONSE: We thank the reviewer for pointing this out. The absence of Ago2 impairs the siRNA pathway, which renders hyper-infection in mosquitoes, but also significantly impacts DNA repair, autophagy and histone gene expression in uninfected mosquitoes. Therefore, our data support that Ago2 knockout itself could impair DNA repair, autophagy and histone abundance, however, virus infection strengthens these impacts. We agree that knockdown of histone globally affects the gene expression, including the autophagy genes (**Fig.6j** in manuscript) and DNA repair genes (as shown in the **Revised Fig.9a, see below in response to reviewer 3**). This was discussed in line 500-510.

Other comments:

COMMENT 3. The results that are presented in Figure 1 are not described well in the text. To which part of Figure 1 do lines 122-126 refer to (Extended Figure 1F is irrelevant)?

RESPONSE: We thank the reviewer for pointing this out. We have updated the presentation of **Fig.1**.

COMMENT 4. In Fig. 1g, the number of samples of *Ago2*^{-/-} mosquitoes in the graph that measures pfu/mL seems much lower than indicated in the graph next to it (infected %). In general, the samples of *Ago2*^{-/-} mosquitoes are much smaller than WT mosquitoes.

RESPONSE: The number of both *Ago2*^{-/-} mutants and WT mosquitoes are indicated on graph (infected %), and the sample size is very similar. The low number of samples of *Ago2*^{-/-} mutants for virus titration in **Fig.1g** is because there were only a few samples collected from *Ago2*^{-/-} mutants positive for viruses.

COMMENT 5. In Fig.1e and 1f, it should be indicated that the midgut epithelium is infected. In Extended Fig. 1d, other tissues besides midgut epithelium (tracheae, muscle?) need to be indicated.

RESPONSE: We indicated infected tracheas (white arrows) in **Extended Fig.1d (now Extended Fig.3)**. We did not indicate midgut epithelium cells in **Fig.1e,f** because we present low magnification of these two figures to show overall infection patterns of midguts. It is difficult to indicate single cells in these figures, but we present high magnification of images in **Fig.3b,c**.

COMMENT 6. Line 146: “evenly distributed” does not seem to be correct as there is a clear peak of 21 nt (Fig 2b). What is clear is that the “background” reads of the small RNAs is considerably higher in *Ago2*^{-/-} than in WT – this is a clear difference (Fig 2b). “Evenly distributed” applies to Fig. 2e, not 2b.

RESPONSE: We corrected it and rephrased this part. The high “background” reads of the small RNAs (non 21-nt siRNA reads) are produced from the positive strand of the MAYV genome (**Extended Data Fig.4b**).

COMMENT 7. When the authors talk about vpiRNAs, they should describe how the vpiRNAs were determined. On the basis of length, it cannot be decided. In that case, the term vpiRNA-like sRNAs should be used but of course it is important to determine to what extent bona fide vpiRNAs are produced (e.g. ping-pong amplification).

RESPONSE: Thank you for pointing this out and we remove the result of vpiRNAs since the manuscript mainly focuses on siRNA pathway.

COMMENT 8. In Extended Fig. 2c, is the scale in the Y-axis correct? Why is there no peak at 21 nt?

RESPONSE: We thank the reviewer for this observation. We found 21nt peak in WT and *AgoN*^{-/-} mosquitoes only at 7 days post DENV2 infection. At 2 days post infection, viral sRNAs represent RNAi-mediated degradation of ingested virus as reported by Franz et al. 2006 and Hess et al. 2011, therefore there is no peak at 21 nt, which is consistent with results presented in Fig.3f from Olmo et al. 2018 and Figure 2A from Hess et al. 2011. Recent results from *Dcr2*^{-/-} mutants showed that 23 to 30 nt viral sRNAs were products of the piRNA pathway (Samuel et al. 2023).

References

Franz AW, Sanchez-Vargas I, Adelman ZN, et al. Engineering RNA interference-based resistance to dengue virus type 2 in genetically modified *Aedes aegypti*. *Proc Natl Acad Sci U S A*. 2006;103(11):4198-4203.

Hess AM, Prasad AN, Ptitsyn A, et al. Small RNA profiling of Dengue virus-mosquito interactions implicates the PIWI RNA pathway in anti-viral defense. *BMC Microbiol*. 2011;11:45.

Olmo RP, Ferreira AGA, Izidoro-Toledo TC, et al. Control of dengue virus in the midgut of *Aedes aegypti* by ectopic expression of the dsRNA-binding protein Loqs2. *Nat Microbiol*. 2018;3(12):1385-1393.

Samuel GH, Pohlenz T, Dong Y, et al. RNA interference is essential to modulating the pathogenesis of mosquito-borne viruses in the yellow fever mosquito *Aedes aegypti*. *Proc Natl Acad Sci U S A*. 2023;120(11):e2213701120.

COMMENT 9. Line 203: “strong fluorescence with a “ball”-like shape”: the ball-like shape have midgut cells, not the midgut?

RESPONSE: The “ball”-like shape has multiple midgut cells as shown in **Fig.3b**, and we added this information in the revised manuscript.

COMMENT 10. Line 220: “clusters of virions were observed in the midguts” should be “in the midgut cells”?

RESPONSE: We have corrected this.

COMMENT 11. Staining of midgut epithelial cells: only staining for viral antigen and DAPI is shown. It would be useful to stain for typical markers in the epithelial cells to reveal cell shape and cell identity.

RESPONSE: We agree that it would be useful if we could stain the midgut epithelium cells, but there are no typical markers specially for *Ae. aegypti* midgut epithelial cells.

COMMENT 12. Line 313: “These data indicate that either silencing of histone mRNAs or inhibiting histone modification adversely affects arbovirus replication in the mosquitoes and renders them more susceptible to arboviral infection.”: re-formulate; silencing of histones results in a decrease in virus titer.

RESPONSE: We have rephased it as suggested.

COMMENT 13. Line 383: “mortality in *Dicer2*^{-/-} mutants is lower than that in *Ago2*^{-/-} mutants”: provide quantitative comparison.

RESPONSE: We have rephased this part of discussion.

COMMENT 14. Line 406: “other conserved immune signaling cascades, including the Toll and apoptosis pathways, are induced upon virus infection and may play a role in defending against viral infection”: the virus may not be the primary trigger of the immune cascades but instead the damage inflicted by uncontrolled virus replication.

RESPONSE: We thank the reviewer for bringing these to our attention. In mosquitoes, the immune cascades are induced by infection with bacteria, viruses, fungi and parasites, and injury may also induce immune response. We have rephased this sentence as “Here we also show that when the siRNA pathway is impaired, other conserved immune signaling cascades, including the Toll and apoptosis pathways, are induced upon virus infection or other stresses and injuries and may play a role in defending against viral infection.”

COMMENT 15. Line 417: “*Ago2* disruption does not affect the endonuclease activity of *Dcr2*”: this is also known in *Drosophila* (provide reference).

RESPONSE: We have updated this sentence and added references.

REVIEWER #3 (Remarks to the Author):

In the present manuscript by Dong and Dimopoulos, the authors describe a new *Ago-2* mutated mosquito model and its impacts on arbovirus infection and mosquito survival after infection. The manuscript includes a lot of valuable information and a new gene knockout model to study various antiviral responses and mechanisms, but there are some major concerns that need to be addressed in this reviewer’s opinion.

COMMENT 1. The authors describe the *Ago-2* mutant that has a 72 bp deletion as an *Ago2* knockout mosquito. I am not sure if this is an entirely fair representation. It is a deletion mutant, lacking part of the protein, but if the rest of the protein is still made, I am hesitant to accept the term ‘knockout’.

RESPONSE: The deletion of 72 bp is in *Piwi_Ago*-like domain that is essential for the *Ago2* function. To confirm the rest of the protein still has *Ago2* function, we compared dsRNA silencing of endogenous genes in WT and *Ago2*^{-/-} mutants. Our results showed a 72 bp deletion in *Piwi_Ago*-like domain leads to loss-of-function of *Ago2* in respect to RNA silencing in *Ae. aegypti*.

COMMENT 2. The authors do not show a dysfunction in the RNAi pathway using for example dsRNA silencing of an endogenous gene. Now with the relevant deletion in the *Ago* domain, a lack of slicing activity is implied, but evidence of it would have been good to show (e.g. parallel injections of wt and mutant mosquitoes with dsRNA, validation of silencing in wt, lack of silencing in mutant)

RESPONSE: As also suggested by reviewer 1, we selected three genes (*AeIMPDH*, *AeCRVP* and *AeTry196*) that have been previously reported to be efficiently silenced by dsRNA in wild type mosquitoes (Angleró-Rodríguez et al. 2017 and Tikhe et al. 2022). These genes did not show silencing in *Ago2*^{-/-} mutants as compared to the WT (**Revised Fig.1c, see above in responses to reviewer 1**). We also silenced *EOF1* (eggshell organizing factor 1, AAEL012336) that significantly impacts fecundity (Isoe et al. 2019) in different species of mosquitoes, and our results confirmed that silencing of *EOF1* significantly reduced fecundity in WT females but did not have effect on fecundity in *Ago2*^{-/-} mutants (**Revised Fig.1b, see above in responses to reviewer 1**), demonstrating siRNA pathway is dysfunctional in *Ago2*^{-/-} mutants.

References

Angleró-Rodríguez YI, Talyuli OA, Blumberg BJ, et al. An *Aedes aegypti*-associated fungus increases susceptibility to dengue virus by modulating gut trypsin activity. *Elife*. 2017;6:e28844.

COMMENT 3. It is hard to get overly excited about the histone/DNA repair/autophagy axis considering the immense impact that high levels of replication appear to have on the cells. Is histone downregulation truly the ‘first’ issue or is it merely a by-product of impaired DDR/other cellular damage? The causes and consequences of Ago2 loss of function and increased virus replication are hard to discern for sure from the present study.

RESPONSE: To address whether histone downregulation is a ‘primary’ (or “first”) issue or a by-product of impaired DNA damage response (DDR), we investigated effects of histone silencing on expression of DNA repair genes, and vice versa, in wild type *Ae. aegypti*. Our results show that silencing of histone genes significantly affects all the DNA repair genes at 3 days post injection (dpi) of dsRNA and 7 of 8 DNA repair genes at 7 dpi (**Revised Fig.9a,b, see above**). Silencing of DNA repair genes also reduces mRNA abundance of *His3* at 2 dpi but does not affect mRNA abundance of all three histone genes at 7 dpi even though silencing of DNA repair genes has occurred (**Revised Fig.9c,d, see above**). Then, we silenced individual DNA repair genes and investigated the impact on expression of histone genes. We found that silencing of *cdk1* had the most effect on mRNA abundance of histone genes at 3 dpi (**Revised Fig.9e, see above**). At 7 dpi, gene silencing was only persisting in *dsRAD50*-injected mosquitoes, but its silencing increased expression of *His2A* and *His3* (**Revised Fig.9f, see above**). Therefore, effect of silencing of DNA repair genes on histone expression is inconclusive. Taken together, our data support that histone downregulation impairs DDR as compared to the effect of DDR silencing on histone gene expression. See figure above.

The plaque assay data (**Fig.1d** in manuscript) in WT and *Ago2*^{-/-} mutants strongly support increased virus titer is a consequence of *Ago2* loss of function.

COMMENT 4. There were a lot of discrepancies with reference to figures and content of figures as described below that made it hard to follow at times. Please proof-read extensively.

RESPONSE: We have made corrections accordingly.

Specific comments:

COMMENT 5. Line 126: In the files provided to this reviewer, no 'Extended Figure 1f' was present. Extended Figure 1 ends with Extended Figure 1D. The paper also does not reference any Extended Figure 1E. Please double check what happened there.

Line 129: Ok you then reference Figure 1g for the count of female mosquitoes which fed, so I assume the previous reference to Extended Figure 1D, was intended for Figure 1g. Seems like the reference to 1h also needs to be to 1i and the one currently saying 1i needs to be 1j. Please fix these.

RESPONSE: We thank the reviewer for pointing this out. We have updated the presentation of Fig.1.

COMMENT 6. Figure Legend 1H – 'percentage of mosquitoes in each cup' – without further context, this is not a sufficient legend and it is unclear what is shown unless you find the relevant section in the text. It may not be required, but it would be ideal if the reader can understand the figure+legend on its own.

RESPONSE: We have revised this.

COMMENT 7. Line 159: The authors state that 'Therefore, our results show that *Ago2* disruption does not affect the production of arboviral sRNAs but does impair the ability of the siRNA pathway to degrade viral RNAs.' Where do the authors show that degradation of viral RNA is impaired? The authors show an increase in viral RNA and a similar abundance of sRNAs (relatively per genome copy), but how can the authors conclude that the sRNA response is inhibited at this stage? (Aside from drawing from their hypothesis). While I don't disagree that this is highly likely the case, the data does not provide any evidence at this point. Only a functional assay measuring *Ago-2* slicing activity would provide this direct evidence.

RESPONSE: We thank the reviewer for pointing this out. As suggested, we checked the silencing efficiency of endogenous genes in embryo stage and adult females in both WT and *Ago2*^{-/-} mutants, and the results showed *Ago2*-mediated silencing is fully abolished in *Ago2*^{-/-} mutants (**Revised Fig.1, see above in responses to reviewer 1**), and an increase in viral RNA and particles could be resulted from impairment of the siRNA pathway.

COMMENT 8. Figure 4g ◊ the MAYV infected *Ago2* knockout midguts overlay image appears to be missing the TUNEL channel in the overlay. The individual channels are shown, but the overlay shows no 'red', while this is clearly visible in the wildtype cell overlay.

RESPONSE: We have corrected this accordingly.

COMMENT 9. Line 278 – typo, should be 'involved' with a d at the end

RESPONSE: We have corrected this accordingly.

COMMENT 10. Line 304: The authors state: 'In the WT mosquitoes, MAYV or DENV2 infection significantly reduced the expression of histone genes when compared to non-infected mosquitoes'. For MAYV in Fig 5i, there is no evidence that a direct mock vs MAYV condition showed significantly reduced histone gene expression. All shown significance bars/asterisks span from one mosquito group to the other, not simply between wt mock and wt MAYV. Having said that, the Extended Figure 5b seems to show an impact of MAYV infection on histone 4. However, it is unclear how that figure differs from Figure 5i (experimentally – both say 4 days post MAYV infection. Are they replicates??).

Going back to the original statement in the paper, DENV infection (Extended Figure 5c) if anything increased expression of histone genes in wildtype mosquitoes. Please fix this entire section, the references to figures and make the shown results clearer.

RESPONSE: We have corrected this section and revised our statement. **Fig.5i** and **Extended Figure 5b** (now **Extended Figure 7f**) show the expression of histone genes in carcasses (the mosquito body without midgut) and midguts, respectively. We used one-way ANOVA to compare the expression of MAYV vs mock between WT and *Ago2*^{-/-} mutants,

and the difference between WT and *Ago2*^{-/-} mutants undermines the difference between MAYV and mock. When we used an unpaired t-test, we observed significant difference of *His4* in WT midguts (**Revised Fig.10a**) and *His2A* in WT carcasses (**Revised Fig.10b**) between MAYV and mock. See figure below.

COMMENT 11. Lines 310-315: When the authors introduce the HDAC inhibitor, it might be better to provide a bit more context for a broad audience. The authors simply say ‘inhibiting histone modification adversely affects arbovirus replication’. HDAC inhibition would directly result in an increase in euchromatin formation and prevent compact/strong histone-DNA interaction. If a HAT inhibitor was used, the opposite results may occur. So simply labeling it ‘histone modification’ is a bit simplistic here.

RESPONSE: We have added introduction to the HDAC inhibitor.

COMMENT 12. Figure 6a shows the 3 genes, p16, Atg2 and Atg6, but the text refers to p62 and two Atgs. Please correct/clarify as needed (e.g. was p16 just a typo and it is p62?).

RESPONSE: This was a typo and should be p62. We have corrected it.

COMMENT 13. Line 328: the authors refer to the confirmation of ‘the upregulation of p62 in *Ago2*^{-/-} mutants with or without MAYV infection at 4 dpi’. However, the figure shows that p62 is downregulated in uninfected *Ago2* knockout mosquitoes and then upregulated back to the wt levels following infection. Furthermore the authors refer to changes in these genes (p62, Atgs) in DENV infected mosquitoes in Extended Figure 6a, but extended Figure 6a is a Western Blot of ATG-8 specifically.

RESPONSE: We only observed the downregulation of *p62* in MAYV-infected *Ago2* knockout mosquitoes and rewrote it as “Our results confirmed the downregulation of *Atg* and *P53* in *Ago2*^{-/-} mutants with or without MAYV infection at 4 dpi (**Fig.6b**), and with or without DENV2 infection at 14 dpi (**Extended Data Fig.9b**), as compared to the WT controls. We also observed downregulation of *p62* in uninfected *Ago2*^{-/-} mutants as compared to the WT mosquitoes and MAYV infection increased *p62* expression in *Ago2*^{-/-} mutants (**Fig.6b**).”

COMMENT 14. Line 359, the authors refer to Figure 4J, but it is Figure 6J.

Line 360 – I believe (?) the correct figure to reference here is Extended Figure 6f . Please check all these references and figure panels.

RESPONSE: We have corrected these typos.

COMMENT 15. Line 360-362: This is again slightly inaccurate – the authors state that ‘As compared to the sugarfed mosquitoes, the bloodfed mosquitoes exhibited a significantly higher expression of autophagy genes in the GFP control mosquitoes, but not in the histone-silenced mosquitoes.’ However, in the referenced Figure 6, expression of two of the autophagy genes (*Atg-7* and *Atg-12*) does increase when comparing between the histone-silenced sugarfed mosquitoes and the histone-silenced bloodfed mosquitoes – just not to the same degree as in the GFP control. Please make sure you

are accurate in your descriptions.

RESPONSE: We have revised it as “As compared to the sugarfed mosquitoes, the bloodfed mosquitoes exhibited a significantly higher expression of *Atg5* and *Atg10* in the GFP control mosquitoes, but not in the *histone*-silenced mosquitoes.”

COMMENT 16. The model in Figure 7 is labeled ‘miRNA’ as the small RNA in Ago-2 ... this should obviously be siRNA.

RESPONSE: We have corrected this accordingly.

REVIEWERS' COMMENTS

Reviewer #1 (Remarks to the Author):

The authors have fully addressed all my concerns and significantly improved the manuscript. I judge the data to be sufficiently sound and in support of the central claims of the manuscript. Therefore, I deem the new version of the manuscript suitable for publication and consider this study a significant contribution to our understanding of mosquito biology.

Reviewer #2 (Remarks to the Author):

The authors did a lot of effort to revise the manuscript (additional experiments, more careful interpretation of data) according to the comments of the reviewers and the quality has improved significantly. I would like to suggest that the authors re-read the manuscript carefully such that typos and wrongly attributed references are avoided.

Reviewer #3 (Remarks to the Author):

Overall this reviewer is ok with most of the revisions. I appreciate the effort that went into the experiments in revised (extended data) Figure 1 to show a clear lack of RNAi capability of these mosquitoes. However, Extended Data Figure 1c. is very hard to interpret and unclear. While this reviewer appreciates the additional effort of generating the data shown in the revised Figure 1c, what is this 'Fold change'? In a traditional silencing experiments, the fold change on the y-axis would be in comparison to the GFP control, meaning that the GFP control would be normalized to a value of 1 and the reader can see the increase/decrease of the gene-specific group easily, for example as a 0.5 fold change compared to GFP. Why is the 'fold change' for these genes lower in the Ago2^{-/-} GFP control versus the WT GFP control? How was the fold change normalized and why are there significant differences in the controls? The methods state that the $2^{-\Delta CT}$ or $2^{-\Delta\Delta CT}$ method was used, but how can it be 'or' and when using $2^{-\Delta\Delta CT}$, the control should have a value close to 1 (?). Please clarify and/or fix data presentation.

The provided revised/extended data Figure 2 is nice and informative. I would personally prefer if it were a main text figure (as should revised Figure 1 in my opinion), but as long as it is included, that's ok.

Revised Figure 3 cannot be found in the revised manuscript (?). I do not see these images in the main text or extended data panels (?). While revised Figure 3 appears to show no 'lysis' as the authors say, there appear to be clear differences in DAPI organization in the midgut. Staining within the nuclei appears more diffuse in Ago2^{-/-} which may indicate disruption in DNA chromosomal integrity. Please also use the magnification for each image comparison – wild-type ovaries were imaged at different scales as Ago2^{-/-} ovaries based on the differences in scale bars, making comparison harder.

Revised Figure 6 appears to be the identical graph to Revised Figure 5b – something went wrong here... (Figure 5b is the one that's wrong, I believe). Having said that, it appears that this analysis (Figure 5) was only for reviewer 1 and is not included in the manuscript either way.

A lot of additional data was generated to solidify or adjust the conclusions from this paper, which is appreciated.

RESPONSES TO REVIEWER COMMENTS

Reviewer #1 (Remarks to the Author):

COMMENT: The authors have fully addressed all my concerns and significantly improved the manuscript. I judge the data to be sufficiently sound and in support of the central claims of the manuscript. Therefore, I deem the new version of the manuscript suitable for publication and consider this study a significant contribution to our understanding of mosquito biology.

RESPONSE: We thank the reviewer for positive comment and encouragement.

Reviewer #2 (Remarks to the Author):

COMMENT: The authors did a lot of effort to revise the manuscript (additional experiments, more careful interpretation of data) according to the comments of the reviewers and the quality has improved significantly. I would like to suggest that the authors re-read the manuscript carefully such that typos and wrongly attributed references are avoided.

RESPONSE: We appreciate the reviewer's comments. We corrected some typos and checked references carefully.

Reviewer #3 (Remarks to the Author):

COMMENT 1: Overall this reviewer is ok with most of the revisions. I appreciate the effort that went into the experiments in revised (extended data) Figure 1 to show a clear lack of RNAi capability of these mosquitoes. However, Extended Data Figure 1c. is very hard to interpret and unclear. While this reviewer appreciates the additional effort of generating the data shown in the revised Figure 1c, what is this 'Fold change'? In a traditional silencing experiments, the fold change on the y-axis would be in comparison to the GFP control, meaning that the GFP control would be normalized to a value of 1 and the reader can see the increase/decrease of the gene-specific group easily, for example as a 0.5 fold change compared to GFP. Why is the 'fold change' for these genes lower in the *Ago2*^{-/-} GFP control versus the WT GFP control? How was the fold change normalized and why are there significant differences in the controls? The methods state that the $2^{-\Delta\Delta CT}$ or $2^{-\Delta\Delta CT}$ method was used, but how can it be 'or' and when using $2^{-\Delta\Delta CT}$, the control should have a value close to 1 (?). Please clarify and/or fix data presentation.

RESPONSE: We appreciate the reviewer's comments. We used the $2^{-\Delta\Delta CT}$ method to normalize the gene expression difference between samples. In each experiment, we had four groups of samples, dsRNA of the target gene-injected WT and *Ago2*^{-/-} mutants and dsRNA of GFP-injected WT and *Ago2*^{-/-} mutants. We changed our data presentation as suggested, and CT values from all the samples were normalized to dsRNA of GFP-injected WT. The reason why the 'fold change' for these genes is lower in the *Ago2*^{-/-} GFP control versus the WT GFP control is that the expression of these genes is lower in the *Ago2*^{-/-} mutants. When using $2^{-\Delta\Delta CT}$, we normalize CT values from all the samples to one sample.

Because each group has four samples, the mean CT value of the control samples will have a value close to 1.

COMMENT 2: The provided revised/extended data Figure 2 is nice and informative. I would personally prefer if it were a main text figure (as should 'revised Figure 1 in my opinion), but as long as it is included, that's ok.

RESPONSE: We appreciate the reviewer's suggestions, but we keep these two figures as supplementary figures since the manuscript mainly focuses on arbovirus infection-induced mortality.

COMMENT 3: Revised Figure 3 cannot be found in the revised manuscript (?). I do not see these images in the main text or extended data panels (?). While revised Figure 3 appears to show no 'lysis' as the authors say, there appear to be clear differences in DAPI organization in the midgut. Staining within the nuclei appears more diffuse in *Ago2*^{-/-} which may indicate disruption in DNA chromosomal integrity. Please also use the magnification for each image comparison – wild-type ovaries were imaged at different scales as *Ago2*^{-/-} ovaries based on the differences in scale bars, making comparison harder.

RESPONSE: We appreciate the reviewer's careful observation. As suggested, we included this figure in the manuscript as a supplementary figure (Supplementary Fig.5e). It may have differences in DAPI organization in the midgut cells in *Ago2*^{-/-} mutants as compared to the WT midgut, but these cells were not shown lysis as we observed in arboviruses-infected midguts of *Ago2*^{-/-} mutants. We removed the ovary IFA images to keep consistent with arboviruses-infected tissues since we only showed midgut and salivary gland IFA images in arboviruses-infected mosquitoes.

COMMENT 4: Revised Figure 6 appears to be the identical graph to Revised Figure 5b – something went wrong here.... (Figure 5b is the one that's wrong, I believe). Having said that, it appears that this analysis (Figure 5) was only for reviewer 1 and is not included in the manuscript either way.

RESPONSE: We appreciate the reviewer's careful observation. There is a duplicate of Revised Figure 5b and Revised Figure 6. We fixed that and presented a new Revised Figure 6 (see below).

COMMENT 5: A lot of additional data was generated to solidify or adjust the conclusions from this paper, which is appreciated.

RESPONSE: We appreciate the reviewer's positive comments.